PREPARED FOR SUBMISSION TO JHEP

YITP-23-59

# Solitonic symmetry as non-invertible symmetry: cohomology theories with TQFT coefficients

Shi Chen,[1] Yuya Tanizaki[2]

[1]*Department of Physics, The University of Tokyo, 7-3-1 Hongo, Bunkyo-ku, Tokyo 113-0033, Japan*

[2]*Yukawa Institute for Theoretical Physics, Kyoto University, Kitashirakawa Oiwakecho, Sakyo-ku, Kyoto 606-8502, Japan*

*E-mail:* s.chern@nt.phys.s.u-tokyo.ac.jp, yuya.tanizaki@yukawa.kyoto-u.ac.jp

ABSTRACT: Originating from the topology of the path-integral target space $Y$, solitonic symmetry describes the conservation law of topological solitons and the selection rule of defect operators. As Ref. [1] exemplifies, the conventional treatment of solitonic symmetry as an invertible symmetry based on homotopy groups is inappropriate. In this paper, we develop a systematic framework to treat solitonic symmetries as non-invertible generalized symmetries. We propose that the non-invertible solitonic symmetries are generated by the partition functions of auxiliary topological quantum field theories (TQFTs) coupled with the target space $Y$. We then understand solitonic symmetries as non-invertible cohomology theories on $Y$ with TQFT coefficients. This perspective enables us to identify the invertible solitonic subsymmetries and also clarifies the topological origin of the non-invertibility in solitonic symmetry. We finally discuss how solitonic symmetry relies on and goes beyond the conventional wisdom of homotopy groups. This paper is aimed at a tentative general framework for solitonic symmetry, serving as a starting point for future developments.

# 1  Introduction

Symmetry provides one of the guiding principles when studying strongly-coupled physics in quantum field theories (QFTs), and astonishingly, the notion of symmetry itself has been vastly generalized in these recent years. The generalization is achieved under the motto that the topologicalness of operators should always represent conservation laws, and we identify the algebraic structure of topological operators in QFTs with the generalized symmetries. Such generalizations mainly include two directions: One is the higher-form symmetry [2] and higher-group symmetry [3–9], where the symmetry operators are defined on various codimensions, and the more recent one is the non-invertible symmetry, where the fusion rule obeys a suitable algebraic structure beyond the usual group multiplications. The non-invertible symmetries in $(1 + 1)$-dim are now well-understood, and the fusion category captures their algebraic structure (when finitely generated) [10–24]. The non-invertible symmetries in higher dimensions start to be realized in various QFTs [1, 25–45], and there is also a massive endeavor to identify the precise mathematical structures behind generalized symmetries [46–60].

In this paper, we shall investigate non-invertible symmetries organizing conservation laws of topological solitons, which we call (non-invertible) solitonic symmetries [1]. Topological solitons are nonperturbative objects in QFTs that appear due to the nontrivial topology of the path-integral target space, such as kinks, vortices, monopoles, etc. They are created/annihilated by defect operators. It has been common wisdom that their topological stability, an intriguing aspect of topological solitons, is captured by the homotopy group of the target space [61–63]. In the previous study [1], the present authors revealed that solitonic symmetry also becomes a non-invertible symmetry in general, and the solitonic symmetry generators are given by the partition functions of auxiliary lower-dim topological QFTs (TQFTs) coupled with the original system. This finding provides us an opportunity to reconsider the foundations of solitonic symmetries, and, surprisingly, deep mathematics turns out to be there behind the solitonic symmetries, which echoes those behind classifying gapped phases [64–70].

The fact that solitonic symmetries become non-invertible implies that the usual homotopy group of the target space $Y$ does not always correctly characterize the topological conservation laws of solitons, which raises the following questions:

- What is the new foundation for solitonic symmetry? (Sec. 2)

- What are the topological operators for solitonic symmetry? (Secs. 3)

- What is the algebraic structure of solitonic symmetry? (Sec. 4)

- What makes solitonic symmetry go beyond homotopy groups? (Sec. 5)

We give proposals and/or solutions to these questions in the indicated sections, and let us summarize the results as follows.

When discussing symmetries, we need to specify the symmetry generators and their action on charged objects. We showed in Sec. 2 that, in the case of the topological con-

servation law of solitons, the charged objects are given by the defect operators that create/annihilate solitons, and then the symmetry generators should be given by topological functionals using the fundamental fields in the path-integral formulation. Then in Sec. 3, we would like to find the most general form of topological functionals to define the non-invertible solitonic symmetries, and we clarify the physical requirements to be satisfied by the topological functionals. In particular, it turns out that the essential requirement comes from locality. As a natural ansatz satisfying such requirements, we propose that the solitonic symmetries are generated by partition functions of auxiliary fully-extended TQFT coupled to the fundamental fields (Ansatz 3.2).

The rigorous treatment of Ansatz 3.2 requires us to employ the knowledge of fully-extended TQFTs. We concisely review the relevant mathematical treatments in Sec. 4. Also in that section, we clarify the algebraic structure of solitonic symmetry, which is described by symmetric fusion higher-categories $\mathsf{Rep}^\bullet(Y)$ for the bosonic case and $\mathsf{sRep}^\bullet(Y)$ for the fermionic case. These observations are consistent with the latest progress on generalized symmetry in the literature, such as Refs. [52–55]. We see that solitonic symmetry serves as non-invertible generalizations of cohomology theories with TQFT coefficients. We also discuss the invertible solitonic subsymmetry given by orthodox cohomology theories.

Armed with this mathematical guidance, in Sec. 5, we can systematically study the origin of non-invertibility in the solitonic symmetry and how the conventional wisdom of homotopy groups is surpassed. We shall see that $(\mathsf{s})\mathsf{Rep}^\bullet(Y)$ can be decomposed into two parts. The first part comes from $(\mathsf{s})\mathsf{Rep}^{\bullet-1}(Y)$ via formulating condensations and contains topological functionals that are trivial on spheres. The second part comes from the homotopy group $\pi_\bullet-$ and contains topological functionals that are nontrivial on spheres. But according to the topological data present in the theory, these spherically-nontrivial topological functionals have to take a non-invertible fusion rule. This decomposition unpacks the structure of solitonic structure inductively and provides us with insight into the connection between the generalized solitonic symmetry from the contemporary perspective and the conventional wisdom since Coleman etc.

## Acknowledgments

This work was initiated during S. C.'s visit to YITP with the Atom-type visiting program, and the authors appreciate the hospitality of Yukawa Institute. The authors are grateful to Mayuko Yamashita for bringing Ref. [68] to their scope in a seminar. They also thank Kantaro Ohmori for valuable comments on the manuscript. S. C. thanks Yuji Tachikawa for cheerful discussions and useful comments on the idea developed in this paper. This work was supported by JSPS KAKENHI Grant No. 21J20877 (S. C.), 22H01218 and 20K22350 (Y. T.), and also by the Center for Gravitational Physics and Quantum Information (CGPQI).

## 2 Basic concepts

Solitonic symmetry is generated by topological functionals. It (i) describes the conservation law in the solitonic sector, and (ii) prescribes the selection rule in correlation functions between solitonic defects, and (iii) determines the possible topological couplings with a background gauge field. We explain these basic notions in this section. The spacetime dimension is denoted by $d$ throughout.

### 2.1 Target space of the path integral

Let us consider a general situation where the quantum field theory (QFT) is defined by the path integral, and its partition function is given by

$$\mathcal{Z} = \int \mathcal{D}\sigma \exp(-S[\sigma]), \tag{2.1}$$

and $\sigma$ is some field on the $d$-dim spacetime. We focus on the non-Grassmann sector of this path integral, and thus $\sigma$ might be a scalar field, a gauge field, a higher-form gauge field, or even a combination of them coupled. We note that, except in the simplest case of pure scalar fields, all other fields are not maps to a fixed space.

In this paper, we are interested in a series of nonperturbative phenomena insensitive to continuous deformations of field configurations. Namely, we only care about the deformation classes of field configurations on some closed (smooth) manifold $M$. The actual choice of $M$ depends on specific problems; it might be the spacetime itself, a submanifold like a space slice or a world line, or a virtual submanifold like the normal sphere bundle of a defect operator (see Sec. 2.2.2). Conveniently, we can always find a topological space $Y$ such that there exists a one-to-one correspondence,

$$\text{Deformation classes of field configurations on } M$$
$$\| \tag{2.2}$$
$$\text{Deformation classes of maps from } M \text{ to } Y.$$

We shall refer to this topological space $Y$ as the *(homotopy) target space* of the path integral and abuse the symbol $\sigma$ to denote also the auxiliary maps to the target space, i.e.,

$$\sigma|_M : M \mapsto Y. \tag{2.3}$$

Formally, the set of deformation classes of field configurations are now expressed by the set of homotopy classes of maps to the target space $Y$,

$$[M, Y] \equiv \{f : M \mapsto Y\}/\text{homotopies}. \tag{2.4}$$

Because the dimension of closed manifold $M$ cannot exceed spacetime dimension $d$, the set $[M, Y]$ depends merely on the homotopy $d$-type of $Y$.

**Proposition 2.1** *In a $d$-dim QFT defined by a path integral, the deformation classes of field configurations on any submanifold depend only on the homotopy $d$-type of the target space $Y$.*

Thus, $Y$ is understood up to $(d{+}1)$-connected maps in this paper. Without loss of generality, we can always require $Y$ to be a $d$-aspherical space.

**Definition 2.2** *Topological space $X$ is $n$-aspherical if $\pi_\bullet(X, x) \simeq 0$ for all $\bullet > n$ and $x \in X$.*

Namely, we can always replace a general $Y$ with its $d$-th Postnikov truncation. Nevertheless, in actual practice, despite Proposition 2.1, we often render more topological and even geometrical structures on $Y$ to keep in touch with other physics that are sensitive to continuous deformations of field configurations.

We now construct several common $Y$'s to illustrate the above abstract concepts.

1. $Y \simeq X$ for a $X$-valued scalar field, where $X$ is a topological space.

2. $Y \simeq BG$, the delooping of $G$ (or the classifying space of $G$), for a gauge field with gauge group $G$, where $G$ is a topological group (can be discrete).

3. If we couple the two fields above via a continuous $G$-action on $X$, then $Y$ fits into a fibration $X \to Y \to BG$ determined by the $G$-action[1].

4. $Y \simeq B^p G$, the $p$-th delooping of $G$, for a $p$-form gauge field with Abelian $G$.

5. $Y \simeq B\mathsf{G}$ for a gauge field of a local higher-group symmetry $\mathsf{G}$ (see the discussions around Philosophies 4.1 and 4.2, and also the remark in Sec. 6.1).

Concrete examples will appear later.

## 2.2 Solitonic symmetry

From a contemporary viewpoint of QFTs, the notion of symmetry is vastly generalized and can be summarized as

$$\text{Generalized symmetry} \equiv \text{Algebra of topological operators.} \tag{2.5}$$

In the path-integral formalism, there are basically two constructions of operators[2], defects and functionals. Both types of operators have the chance to be topological. Topological defects are the most orthodox topological operators and all symmetries "on the electric side" are generated by them. Topological functionals are more or less atypical and symmetries "on the magnetic side" are generated by them. In this paper, we shall refer to the symmetry generated by topological functionals as *solitonic symmetry*, i.e.,

$$\text{Solitonic symmetry} \equiv \text{Symmetry generated by topological functionals}. \tag{2.6}$$

The core purpose of this paper is to reveal the structure of solitonic symmetry by studying the behavior of topological functionals.

---

[1] If $G$ acts on $X$ freely (i.e., the action has no fixed point), we just have $Y \simeq X/G$. If $X$ is contractible, we just have $Y \simeq BG$. Otherwise, $Y$ has a quite complicated structure, such as many orbifolding examples.

[2] We note that these two constructions can be interchanged under the duality operation, and thus the notion of the solitonic symmetry depends on the explicit path-integral realization of a given QFT.

### 2.2.1 Elementary properties

For a functional on an $n$-dim closed manifold $M$ to be topological, it must factor through the deformation classes of field configurations. Namely, it factors through $[M, Y]$ for the target space $Y$. Given that $[M, Y]$ depends only on the homotopy $n$-type of $Y$, we see the following property of topological functionals.

**Proposition 2.3** *The $n$-dim topological functionals depend only on the homotopy $n$-type of the target space.*

This is consistent with Proposition 2.1. As a prototypical example of topological functionals, let us pick up a cohomology class $\omega \in \mathbb{E}^\bullet(Y)$ for some multiplicative cohomology $\mathbb{E}$[3]. It can be an ordinary cohomology $\mathbb{H}R$ for some ring $R$ or an extraordinary cohomology such as $\mathbb{K}$ and $\mathbb{K}\mathbb{O}$. Then, on any closed $n$-dim $\mathbb{E}$-orientable manifold $M$ that acquires a chosen $\mathbb{E}$-orientation, we can define a topological functional as

$$U_g(M) \equiv g\left(\int_M \sigma^* \omega\right), \qquad \forall g \in \mathrm{Hom}\big(\mathbb{E}_{n-\bullet}, U(1)\big). \tag{2.7}$$

We can readily see the fusion rule $U_{g_1} U_{g_2} = U_{g_1 g_2}$ and thus we obtain a $p$-form symmetry with $p = d - n - 1$. The operator dimension $n$ can range from $d$ to $0$, corresponding to the symmetry form $p$ ranging from $-1$ to $d - 1$. Note that $n$-dim topological functionals are exactly $\theta$-angles, i.e. topological terms in the action that depend on $Y$ only[4]. We shall see concrete examples of operator (2.7) as soon as in Sec. 3.1.

The solitonic symmetry defined above has an invertible fusion rule, which is captured by an Abelian group, $\mathrm{Hom}(\mathbb{E}_{n-\bullet}, U(1))$ or one of its quotients. Actually, operator (2.7) gives the universal construction of invertible solitonic symmetry. In this paper, we shall discuss the most generalized connotation of topological functional and solitonic symmetry with more complicated non-invertible fusion rules. In particular, non-invertible $\theta$-angles mean couplings to topological orders, as explained in Refs. [1, 71]. Nevertheless, in any case, the fusion rule of topological functionals must still be commutative, because on each supporting manifold, the fusion is just the multiplication of complex numbers:

**Proposition 2.4** *Solitonic symmetry is commutative.*

Namely, topological functionals never care about their order. Solitonic symmetry is also insensitive to the theory details, including the action and ambient spacetime, because topological functionals and their fusions do not care about these theory details. Therefore, when a $d$-dim and a $(d+1)$-dim theory share the same target space $Y$ (i.e., the homotopy $d$-types of the target spaces are identical), we have for $0 \leq n \leq d$,

$$n\text{-dim topological functional in } d\text{-dim QFT}$$
$$\| \tag{2.8}$$
$$n\text{-dim topological functional in } (d+1)\text{-dim QFT},$$

---

[3]Spectrum $\mathbb{E}$ can be assumed connective, i.e. $\mathbb{E}_\bullet \simeq 0$ for $\bullet < 0$, given Proposition 2.3.

[4] In contrast, topological terms that also depend on the geometry beyond the mere topology of $Y$ are not $n$-dim topological functionals. CS terms and WZW terms are such counter-examples.

and accordingly, for $d-1 \geq p \geq -1$, we have the following equivalence,

$$p\text{-form solitonic symmetry in } d\text{-dim QFT}$$
$$\|$$
$$(p+1)\text{-form solitonic symmetry in } (d+1)\text{-dim QFT}\,.$$

(2.9)

From this point of view, the algebraic structure of solitonic symmetry is supposed to be a sort of cohomology theory on the target space $Y$; we shall justify this in Sec. 4.3. Thus we shall neglect to mention spacetime when discussing topological functionals. The spacetime dimension implicitly enters as an upper limit for the possible dimension of topological functionals, according to Prop. 2.1.

Before continuing the journey to more details about topological functionals, let us pause here to acquaint readers with solitonic symmetry's physical consequences.

### 2.2.2 Physical significance

First, a symmetry shows the presence of certain conserved charges. To find charged objects of solitonic symmetry, let us consider a correlation function involving a topological functional. Then this topological functional can produce a nontrivial number as long as the field configuration on its supporting manifold cannot be continuously deformed to a trivial configuration. This can be achieved only if the correlation function includes proper *solitonic defects* or the spacetime has a special topology.

Let us now introduce the notion of solitonic defects. For some $0 \leq p \leq d-1$, we take a $p$-dim submanifold $N$ of the spacetime and excise its infinitesimal neighborhood. Then a $p$-dim Dirichlet defect operator on $N$, which we call a solitonic defect operator, is defined by putting the Dirichlet boundary condition on the boundary of the excised region. More formally, it is a Dirichlet boundary condition on the normal sphere bundle $\mathcal{S}N$ of $N$ (Locally, $\mathcal{S}N \simeq N \times S^{d-p-1}$). It can be virtually viewed as a $(d-1)$-dim submanifold in the spacetime via a tubular neighborhood. According to Sec. 2.1, the deformation classes of such Dirichlet boundary conditions can be expressed by deformation classes of maps

$$\sigma|_{\mathcal{S}N} : \mathcal{S}N \mapsto Y. \tag{2.10}$$

Namely, the deformation classes of solitonic defects on $N$ are classified by

$$[\mathcal{S}N, Y]\,. \tag{2.11}$$

Solitonic defects can be either topological or non-topological operators, depending on the details of the theory. If topological, they also generate symmetry, but "on the electric side" and thus non-solitonic.

Solitonic defects couple to a nonperturbative sector called the *solitonic sector* which appears because of the nontrivial topology of the target space[5]. When we put solitonic

---

[5] The solitonic sector discussed in this paper can exist on closed spacetimes. There are also other types of solitonic sectors that inhabit non-closed spacetimes with boundaries only. They also originate from the path-integral topology but are not captured by the target space. Nevertheless, in many cases, they can still be understood from the target space of another related theory; See Sec. 6.1 for more discussions.

defects in the spacetime and evaluate the correlation function via the path integral, the configurations to be integrated are topological solitons bounded by those defects (see Sec. 2.2.3 for a few examples). In particular, the tree-level contribution to the correlation functions comes from solitonic solutions of the classical equation of motion. Therefore, the solitonic sector in quantum theory can be viewed as the quantization of classical solitons, and the solitonic defects are the creation/annihilation operators for quantum solitons. Aside from putting solitonic defects, arranging a nontrivial spacetime topology is also a common method to visualize the solitonic sector.

It is now clear that the charged objects of solitonic symmetry are exactly those solitonic objects introduced above. Solitonic symmetry puts conserved charges on these solitonic objects, which we call *topological charges*. The conservation law of topological charges constrains the correlation functions among solitonic defects and prescribes the selection rule in the physical processes in which topological solitons are involved.

Second, a symmetry prescribes the possible couplings to background gauge fields. This coupling is topological for solitonic symmetry, namely a topological term in the action. Typically, the coupling is directly related to the solitonic sector. However, some couplings are not related to any authentic solitonic sector. For example, a U(1) Chern-Simons path integral

$$\int \mathcal{D}a \, \exp\left\{ \frac{\mathrm{i}k}{4\pi} \int a\mathrm{d}a \right\} \tag{2.12}$$

has no physical solitonic sector since monopoles are not gauge invariant as point-like local operators. However, we can still couple a $U(1)$ background gauge field $A$ topologically via

$$\int \mathcal{D}a \, \exp\left\{ \frac{\mathrm{i}k}{4\pi} \int a\mathrm{d}a + \frac{\mathrm{i}}{2\pi} \int a\mathrm{d}A \right\}. \tag{2.13}$$

The information of the existence of such possible topological couplings with a background $U(1)$ gauge field is also encoded in a 0-form $U(1)$ solitonic symmetry from $Y \simeq BU(1)$, although the corresponding solitonic objects are unphysical.

Another general case is that, due to dimensional reasons, no solitonic defect carries a $(-1)$-form topological charge. Instantons, the charged objects under $(-1)$-form solitonic symmetry, are just classical objects. Thus $(-1)$-form solitonic symmetry, generated by $\theta$-angles, never really rules a quantum solitonic sector but instead prescribes the topological couplings to a background "0-form gauge field", i.e., a background axion field.

### 2.2.3 Conventional wisdom and homotopy groups

In the conventional discussion on topological solitons, their stability and the selection rules are usually discussed using the homotopy group of the target space. To be concrete, let us consider two simple examples:

(1) An $S^1$ sigma model has a $(d-2)$-dim solitonic defect, on which a kink can end. Their $(d-2)$-form topological charge is related to $\pi_1 S^1 \simeq \mathbb{Z}$. This gives rise to a $(d-2)$-form $U(1)$ solitonic symmetry.

(2) A pure U(1) gauge theory or a U(1) Higgs model has a $(d-3)$-dim 't Hooft defect, on which a magnetic flux or a gauge vortex can end. Their $(d-3)$-form topological charge is related to $\pi_2 BU(1) \simeq \mathbb{Z}$. This gives rise to a $(d-3)$-form $U(1)$ solitonic symmetry.

In these examples, the solitons and their conservation laws are controlled by $\pi_\bullet Y$, the homotopy groups of the target space $Y$. This conventional treatment suggested a conventional wisdom that the $p$-form solitonic symmetry is described by

$$\mathrm{Hom}\Big(\pi_{d-p-1}Y, U(1)\Big). \tag{2.14}$$

Indeed, in the two examples above, the relevant topological functionals can be constructed via the universal invertible form (2.7). Thus the solitonic symmetries in these two examples are indeed invertible and described by Eq. (2.14).

We would like to note that the above examples are actually selected rare cases. In general, most of the topological charges prescribed by homotopy groups cannot be detected by the universal invertible construction (2.7), and the solitonic symmetry is not given by Eq. (2.14). The key problem is the following.

- The notion of the dimension of a solitonic configuration is, in general, ambiguous. A generic soliton may be a mixture of solitons of clear dimensions.

This is especially often the case for the solitonic configuration bounded by solitonic defects, which determines the correlation function of these solitonic defects.

- A $p$-dim solitonic defect may be able to create/annihilate not only $(p+1)$-dim solitons but also solitons of all dimensions $< p + 1$.

- A $p$-dim solitonic defect can carry $q$-form topological charges for all $0 \le q \le p$.

This is not surprising: Any conserved $q$-form charge can be carried by operators of dimension $\ge q$, which has already been noticed since the beginning of generalized symmetry [2]. This phenomenon is the starting point that led us to topological charges beyond homotopy groups in the $\mathbb{C}P^1$ model [1], which provides an example of non-invertible solitonic symmetry. The entire Sec. 5 will be devoted to discussing how general solitonic symmetry goes beyond homotopy groups and becomes non-invertible.

## 3   Topological functional

From the contemporary perspective, understanding solitonic symmetry exactly means understanding topological functionals. We shall carefully inspect the notion of topological functionals in this section. At first glance, functional operators look more orthodox and simpler than defect operators. However, we will find that they are much subtler than defects, and several physical requirements vastly constrain a well-behaved notion of topological functionals. Pursuing a well-behaved notion of topological functionals, we will eventually bring ourselves to Ansatz 3.2 which claims that topological functionals are best understood as the partition functions of auxiliary fully-extended TQFTs.

### 3.1 The identity problem

As we have mentioned at the beginning of Sec. 2.2.1, a topological functional on a manifold $M$ has to factor through $[M, Y]$. However, not every $[M, Y] \mapsto \mathbb{C}$ gives rise to topological functionals. A priori, only the topology of $M$ matters for a topological functional. We cannot distinguish the original $M$ and a new $M$ transformed by a self-diffeomorphism $M \xrightarrow{f} M$. Accordingly, let us consider two different configurations $M \xrightarrow{a} Y$ and $M \xrightarrow{b} Y$ such that $f$ transforms one to the other, say $b = f \circ a$. Then, a topological functional we can construct with our bare hands must be blind to the difference between $a$ and $b$. More formally, let us consider the mapping class group $\pi_0 \mathrm{Diff}(M)$, i.e., the group of the isotopy classes of self-diffeomorphisms on $M$. This group acts on $[M, Y]$ as described above. Then, a topological functional should actually factor through the equivalence classes,

$$[M, Y] \Big/ \pi_0 \mathrm{Diff}(M) \,. \tag{3.1}$$

Unfortunately, in most interesting cases, the $\pi_0 \mathrm{Diff}(M)$-action on $[M, Y]$ results in vast degeneracies. Many elements in $[M, Y]$ must be considered identical. We thus call this the identity problem. This problem terribly prevents us from having sufficiently many meaningful topological functionals.

There is one way out, to add some extra structure to $M$. The self-diffeomorphism $f$ may transform this structure into a different structure of the same type. If this happens, we can distinguish $a$ and $b$ by their relative relationship to the extra structure. More formally, now we should consider the mapping class group $\pi_0 \mathrm{Diff}(M, \gamma) \subseteq \pi_0 \mathrm{Diff}(M)$ of isotopy classes of self-diffeomorphisms that preserve the extra structure $\gamma$. Then a topological functional that relies on $\gamma$ should factor through the equivalence classes,

$$[M, Y] \Big/ \pi_0 \mathrm{Diff}(M, \gamma) \,. \tag{3.2}$$

In the consistent practice of physicists, we almost always unconsciously assume some extra structure; recall "$\mathbb{E}$-orientation" for Eq. (2.7). It is precisely the identity problem that motivates our subconscious to do so. Such $\mathbb{E}$-orientations are primary examples of the structure $\gamma$. Thus we shall just call $\gamma$ a generalized orientation. Consequently, a topological functional that can detect sufficiently many deformation classes of field configurations needs to rely on a generalized orientation $\gamma$.

### Example: orientation

For example, let us consider $Y \simeq S^2$ and $M \simeq S^2$. Then we have $[M, Y] \simeq \pi_2(S^2) \simeq \mathbb{Z}$. Let us consider the reflection self-diffeomorphism $f$ defined by $f(\vec{n}) = -\vec{n}$ where we view $S^2 \subseteq \mathbb{R}^3$. This self-diffeomorphism generates

$$\pi_0 \mathrm{Diff}(S^2) \simeq \mathbb{Z}_2 \,. \tag{3.3}$$

Clearly, $f$ acts on $[M, Y]$ via $\mathbb{Z} \to -\mathbb{Z}$. Thus with bare hands, we cannot distinguish two configurations labeled by opposite integers. To break the ice, we note that $M$ is orientable.

Recall that orientations form a $H^0(-;\mathbb{Z}_2)$-torsor. Thus there are two orientations on $M$, $\xi$ and $\xi'$, which are exchanged under the transformation of $f$. That is,

$$\pi_0 \text{Diff}(S^2, \xi) \simeq 0 \,. \tag{3.4}$$

Combining configurations with orientations, we see that for an $n \in \mathbb{Z} \simeq [M, Y]$, $(n, \xi) \overset{f}{\longleftrightarrow} (-n, \xi')$ and $(-n, \xi) \overset{f}{\longleftrightarrow} (n, \xi')$ are distinct from each other and cannot be mixed by $f$. Based on an orientation, we can construct the following topological functional,

$$U_\theta(M) \;\equiv\; \exp\left\{ i\theta \int_M \sigma^* b \right\}, \qquad \forall \theta \in \mathbb{R}/2\pi\mathbb{Z}\,, \tag{3.5}$$

where $b$ denotes the canonical 2-form on $S^2$ that integrates to 1. This operator can distinguish any two elements in $[M, Y]$. We can also recast operator (3.5) into the universal form (2.7) by choosing $\mathbb{E} \simeq \mathbb{H}\mathbb{Z}$. Concretely, in Eq. (2.7), we take $\omega$ as a generator of $H^2(Y; \mathbb{Z}) \simeq \mathbb{Z}$, and $g \in \text{Hom}(\mathbb{Z}, U(1)) \simeq \mathbb{R}/2\pi\mathbb{Z}$.

## Example: spin structure

Besides orientations, other structures may also be needed, such as spin structures. An interesting example was presented in our earlier work [1] (also implicitly in Ref. [71]). We consider $Y \simeq S^2$ and $M \simeq S^2 \times S^1$. With some efforts we can compute

$$[S^2 \times S^1, S^2] \;\simeq\; \left\{ (m, \ell) \,\middle|\, m \in \mathbb{Z},\, \ell \in \mathbb{Z}_{2|m|} \right\}, \tag{3.6}$$

where $\mathbb{Z}_0$ means $\mathbb{Z}$. For a configuration $\sigma$, the $m$ labels $\sigma|_{S^2 \times \{p\}}$ in $\pi_2(S^2) \simeq \mathbb{Z}$ for an arbitrary $p \in S^1$. Now consider the twist diffeomorphism $f$ defined by $f(\vec{n}, t) \equiv (e^{t \hat{\vec{z}} \times} \vec{n}, t)$, where we view $S^2 \subseteq \mathbb{R}^3$ and $S^1 \simeq \mathbb{R}/2\pi\mathbb{Z}$, as well as another diffeomorphism $g$ defined by $g(\vec{n}, t) \equiv (-\vec{n}, -t)$. Both $f$ and $g$ preserve an orientation. Actually, for an orientation $\xi$, they generate

$$\pi_0 \text{Diff}(S^2 \times S^1, \xi) \simeq \mathbb{Z}_2 \times \mathbb{Z}_2\,. \tag{3.7}$$

Both $f$ and $g$ induce an almost double degeneracy on $[M, Y]$: $(m, \ell) \overset{g}{\longleftrightarrow} (-m, \ell)$ while $f$ exchanges $(m, \ell_1)$ and $(m, \ell_2)$ if $2\ell_1 = 2\ell_2$. For example, $f$ transforms $(\vec{n}, t) \mapsto \vec{n}$ into $(\vec{n}, t) \mapsto e^{t \hat{\vec{z}} \times} \vec{n}$, and these two maps belong to different classes $(1, 0)$ and $(1, 1)$ in $[M, Y]$, respectively.

We would like to lift the $f$-degeneracy. We note that $M$ is spinnable. Recall that, on top of a given orientation, spin structures form a $H^1(-;\mathbb{Z}_2)$-torsor. Since $H^1(S^2 \times S^1; \mathbb{Z}_2) \simeq \mathbb{Z}_2$, $M$ have two different spin structures $\rho$ and $\rho'$ on top of an orientation $\xi$. They are exchanged by $f$. That is,

$$\pi_0 \text{Diff}(S^2 \times S^1, \xi, \rho) \simeq \mathbb{Z}_2\,, \tag{3.8}$$

where only the $g$-degeneracy remains. Based on a spin structure, we can construct the following topological functional as a spin Chern-Simons integral,

$$U_k(M) \;\equiv\; \exp\left\{ ik \int_M \frac{\sigma^* a \, d\sigma^* a}{4\pi} \right\}, \tag{3.9}$$

where $k \in \mathbb{Z}$, $k \sim k + 2$, and $a$ denotes the $U(1)$ gauge field on $S^2$ associated with the Hopf fibration $S^1 \to S^3 \to S^2$, which is related to $b$ in the former example (3.5) via $b = \mathrm{d}a/2\pi$. We can also recast operator (3.9) into the universal form (2.7) by choosing $\mathbb{E} \simeq \mathbb{KO}$ since a $\mathbb{KO}$-orientation is exactly a spin structure. Concretely, in Eq. (2.7), we take $\omega$ as a generator of $KO^2(Y) \simeq \mathbb{Z}$, and $g \in \mathrm{Hom}(KO_1, U(1)) \simeq \mathrm{Hom}(\mathbb{Z}_2, U(1))$. The operator (3.9) can lift the almost double degeneracy in $[M, Y]$ caused by $f$. Nevertheless, distinguishing other elements in $[M, Y]$ (except the $g$-degeneracy) requires non-invertible topological functionals based on a spin structure as the present authors showed in Ref. [1], which will also be discussed in Sec. 5.2.2 in this paper.

## 3.2 The coherence problem

The identity problem concerns a topological functional on a single supporting manifold only. An even more severe problem appears if we move from one supporting manifold to another. Namely, for manifolds $M_1 \not\simeq M_2$, given a function on $[M_1, Y]$ and another function on $[M_2, Y]$, how can we tell whether they are just different topological functionals or different realizations of the same topological functional? An incorrect assignment would lead to wrong solitonic physics. We call this the coherence problem.

We can learn hints from the universal construction (2.7) for invertible solitonic symmetry. It is naturally defined on any closed $\mathbb{E}$-orientable $\mathbb{E}$-oriented manifolds. Also, its concrete incarnations in operators (3.5) and (3.9) are automatically defined on any closed oriented orientable manifolds and any closed spin spinnable manifolds, respectively. It is natural to regard the operators on different manifolds but defined by the same expression Eq. (2.7) as the different realizations of the same topological functional.

The above observation suggests that the solution to the coherence problem is to require locality. Recall that defect operators are defined by infinitesimal boundary conditions around the operator (see Sec. 2.2.2). This definition concerns field configurations in the vicinity of each point on the supporting manifold and thus satisfies a good sense of locality. However, a naively defined functional operator may behave quite non-local and might not yield a physically sensible operator. We propose that a functional operator satisfies locality if it is the multiplication of piecewise data localized around each point. The universal invertible construction (2.7) provides the special cases where multiplication is given the "exponentiation" of summation, represented by morphisms to $U(1)$. Furthermore, gluing these local data may require a generalized orientation on supporting manifolds. In summary,

$$\text{Locality} \equiv \text{Multiplying local data with respect to generalized orientation.} \qquad (3.10)$$

A functional satisfying locality automatically renders coherence on a class of manifolds with the prescribed generalized orientation.

A natural subsequent question is whether there is a universal construction of multiplying local data, which goes beyond the exponentiation (2.7) and can capture non-invertible cases. We propose a positive answer: The universal way of multiplying local data is another path integral, and a most general functional that satisfies locality is the partition function

of another fully-extended QFT. Namely, we can consider some auxiliary fields that inhabit the operator manifold only, couple them to the dynamical quantum fields, and perform a path integral of the auxiliary fields. The output of this auxiliary path integral is the partition function of an auxiliary fully-extended QFT that inhabits the operator manifold only. We optimistically assume that all functional operators satisfying locality can be produced by the partition functions of some auxiliary fully-extended QFT. In particular, we require all topological functionals to satisfy locality, i.e., all topological functionals $\mathcal{T}[M, \sigma]$ are assumed to be produced by the partition functions of auxiliary topological fully-extended QFTs (TQFTs) coupled to the target space $Y$:

$$\mathcal{T}[M, \sigma] = \text{TQFT partition function that couples with } \sigma|_M : M \to Y. \qquad (3.11)$$

In particular, this construction includes Eq. (2.7) as its special cases for invertible topological functionals. Also, this construction is consistent with Propositions 2.3 and 2.1. We now flesh out the precise connotation of "TQFT" in our proposal.

### 3.3 Our ansatz

In a specific theory, which generalized orientation the topological functionals rely on should be determined by the theory itself. Therefore, instead of studying topological functionals for an arbitrary generalized orientation, we focus on the most common generalized orientations in this paper. The most fundamental property of a theory that determines the generalized orientation is particle statistics. As might be surprising at first glance, a non-Grassmann path integral can be used to define not only a bosonic QFT that inhabits any oriented spacetime but also a fermionic QFT that inhabits any spin spacetime. These fermionic theories are not bosonic in disguise, i.e., the $\mathbb{Z}_2$-grading $(-)^F$ on states is nontrivial, as long as the action includes proper spin topological terms.

In a bosonic (resp. fermionic) theory, topological functionals are supposed to inhabit oriented (resp. spin) manifolds. This is natural since the theory itself inhabits oriented (resp. spin) spacetime. A more persuasive rationale is that all the solitonic defects (see Sec. 2.2.2), the charged operators of topological functionals, are defined by field configurations on oriented (resp. spin) manifolds. To see this, one notes that the normal sphere bundle $\mathcal{S}N$ of any closed submanifold $N$ in the spacetime naturally inherits an orientation (resp. a spin structure) from the spacetime, even if $N$ itself is not orientable or spinnable. Consequently, the example in Eq. (3.9) cannot exist in bosonic theories. We focus on the elementary cases of bosonic and fermionic theories in the paper while leaving the theories where other interesting generalized orientations[6] are involved to future work.

We now have all the ingredients to formulate an Ansatz for topological functionals, however, unfortunately, given that $Y$ satisfies a finiteness condition.

**Definition 3.1** *Topological space $Y$ is $n$-finite if $\pi_0 Y$ is finite and $\pi_q(Y, y)$ is finite for all $0 \leq q \leq n$ and $y \in Y$.*

---

[6] Such theories appear typically when one wants to take into account (1) unorthodox statistics, (2) discrete spacetime symmetry, (3) mixing between spacetime and internal (non-solitonic) symmetry, and (4) conditions on higher objects than particles.

If $M$ is a closed manifold of dimension $\leq n$, $[M, Y]$ is finite when $Y$ is $n$-finite. $[M, Y]$ has the chance to have infinitely many elements if $Y$ is not $n$-finite. Infinitely many topological charges are a sign of continuous symmetry. The existence of infinitely many elements in $[M, Y]$ and infinitesimal symmetry transformations cause tricky technical troubles. We shall present a systematic treatment of discrete solitonic symmetry but only an approximating treatment of continuous solitonic symmetry.

Let us describe our ansatz for topological functionals responsible for discrete solitonic symmetry. To produce bosonic (resp. fermionic) topological functionals, TQFTs themselves must be bosonic (resp. fermionic). Their partition functions inhabit closed manifolds equipped with a map to the target space $Y$, $\sigma|_M : M \to Y$. To justify "multiplication of local data", these TQFTs must be maximally local, i.e., they should be fully-extended TQFTs, and we reach the following ansatz[7]:

**Ansatz 3.2** *For an $n$-finite space $Y$, an $n$-dim bosonic (resp. fermionic) topological functional to $Y$ is the partition function of an $n$-dim bosonic (resp. fermionic) $Y$-enriched fully-extended TQFT.*

We regard this ansatz as a complete characterization of topological functionals. Based on the physical principle that a QFT is completely determined by its partition functions (in the presence of various kinds of background fields), we make the following conjecture.

**Conjecture 3.3** *For an $n$-finite space $Y$, inequivalent $n$-dim bosonic (resp. fermionic) $Y$-enriched fully-extended TQFTs produce different $n$-dim bosonic (resp. fermionic) topological functionals to $Y$.*

Ansatz 3.2 and Conjecture 3.3 will be the foundation for our analysis of solitonic symmetry in this paper. We note that similar TQFTs often appear in a thriving contemporary theme of physics, the classification of gapped phases. In particular, they are tightly related to the notion of symmetry-protected topological phases and symmetry-enriched topological orders.

To conclude this section, we slightly discuss continuous solitonic symmetry. There are basically two problems. First, although infinitesimal symmetry generators can be unbounded (self-adjoint operators in the invertible case), finite symmetry operators must be bounded (unitary operators in the invertible case), which means we have to require a topological functional $[M, Y] \mapsto \mathbb{C}$ to be bounded. Second, Conjecture 3.3 fails due to the existence of non-semisimple TQFTs which superfluous produce duplicated partition functions as semisimple TQFTs; recall that for group representations $R \not\simeq A \oplus R/A$, we still have $\mathrm{tr}_R = \mathrm{tr}_A + \mathrm{tr}_{R/A} = \mathrm{tr}_{A \oplus R/A}$. This paper does not attempt a systematic treatment of continuous solitonic symmetry. Instead, we shall be satisfied with the discrete subsymmetries of continuous solitonic symmetry. This is realized by considering $n$-finite homotopy quotients of $Y$.

---

[7]When the theory itself is topological, there have been attempts to classify defect operators [72], which is consistent with our Ansatz.

**Definition 3.4** *$Z$ is a homotopy quotient of $Y$ if there is a map $f : Y \mapsto Z$ such that $f_* : \pi_0 Y \to \pi_0 Z$ is surjective and $f_* : \pi_q(Y, y) \to \pi_q(Z, f(y))$ is an epimorphism for all $q > 0$ and $y \in Y$.*

Picking up an $n$-finite homotopy quotient $Z$ of $Y$ and considering topological functionals that factor through $[-, Z]$, we obtain a discrete solitonic subsymmetry. We believe that the colimit of all discrete subsymmetries (see Sec. 4.3.1) leads to an almost faithful approximation to a continuous symmetry, just like approximating $U(1)$ by $\mathbb{Q}/\mathbb{Z} \simeq \bigcup_{n \in \mathbb{N}} \mathbb{Z}_n$. Besides, we shall find it easy to describe the continuous solitonic symmetry directly in some concrete examples.

## 4  Algebraic structure of solitonic symmetry

We are going to reveal the universal algebraic structure of solitonic symmetry based on Ansatz 3.2 and Conjecture 3.3. First in Sec. 4.1, we present a short mathematical preliminary on *higher-categories*. Then in Sec. 4.2, we shall formulate the mathematical notion of fully-extended TQFTs to clarify the accurate connotation of Ansatz 3.2. Finally in Sec. 4.3, we shall discuss the mathematical structure that describes the algebraic structure of solitonic symmetry.

This section aims to establish a more or less rigorous mathematical ground for solitonic symmetry. Thus our expositions might inevitably look more or less abstract. However, readers acquainted with the issue of classifying gapped phases by fully-extended TQFTs will find the expositions familiar and recognize tremendous echoes.

### 4.1  Preliminaries on higher-categories

The most efficient way to formulate fully-extended TQFTs needs the package of higher-categories. The goal here is to overview this nice mathematical package briefly. We are not attempting a self-contained exposition, but instead, we present an oversimplified introduction following Sec. 1.3 of Ref. [64]. Unfamiliar readers could find good entrances in the literature like Refs. [64, 73, 74] to get started into this evolving contemporary discipline.

#### $n$-category

To start with, let us recollect the definition of categories. A category comprises (i) objects, (ii) a set $\mathrm{Hom}(x, y)$ between any two objects $x$ and $y$, and (iii) an associative unital composition map $\mathrm{Hom}(x, y) \times \mathrm{Hom}(y, z) \to \mathrm{Hom}(x, z)$ for any three objects $x$, $y$, and $z$. In particular, the composition map makes $\mathrm{Hom}(x, x)$ a monoid. Elements of $\mathrm{Hom}(x, y)$ are called morphisms. One of the tremendous reasons why categories are useful is that they are regarded as soft instead of rigid. Namely, we regard two categories as "the same" as long as they are equivalent rather than isomorphic. This is similar to algebraic topology, which cares about (weak) homotopy equivalences instead of homeomorphisms.

Generalizing the above definition, we can sketch the notion of $n$-categories by an induction. At the root of this induction, a 0-category is a set, and its elements are called objects or 0-morphisms. The induction then goes as follows. An $n$-category comprises (i) objects,

also called 0-morphisms, (ii) a small $(n-1)$-category $\mathrm{Hom}(x,y)$ between any two objects $x$ and $y$, and (iii) an associative unital composition functor $\mathrm{Hom}(x,y) \times \mathrm{Hom}(y,z) \to \mathrm{Hom}(x,z)$ for any three objects $x$, $y$, and $z$. In particular, the composition functor makes $\mathrm{Hom}(x,x)$ a monoidal $(n-1)$-category. For all $0 \leq p < n$, $p$-morphisms of $\mathrm{Hom}(x,y)$ are called $(p+1)$-morphisms of this $n$-category. Clearly, if we take $n = 1$, we just recover the definition for an ordinary category. We can conceive an $\infty$-category as a proper limit of $n$-categories as $n$ approaches $\infty$. Its $\mathrm{Hom}(x,y)$'s are also $\infty$-categories.

The above definition sketch looks promising but hides the subtlety in treating the associativity and the unitality for compositions. We do not need the too rigid notion of strict $n$-categories, where these conditions are satisfied literally. Instead, we need weak $n$-categories, where these conditions are satisfied up to specified equivalences. For example, in the case of $n = 2$, we want $\mathrm{Hom}(X,X)$ to be a (weak) monoidal category rather than a strict monoidal category. However, as $n$ increases, characterizing accurately all the axioms gets rapidly a formidable task. Different models for organizing this have been proposed, and the equivalence between them, though widely believed, is a matter of ongoing research. We shall drop the prefix "weak" henceforth.

We now introduce two convenient notations. Let us consider an $n$-category $\mathsf{C}$ with a distinguished object $1_{\mathsf{C}}$. For example, $\mathsf{C}$ might be a monoidal $n$-category, which is an $n$-category with the "tensor product" $\otimes$ and the unit object $1_{\mathsf{C}}$. We conceive a monoidal $(n-1)$-category $\Omega\mathsf{C}$ via

$$\Omega\mathsf{C} \equiv \mathrm{Hom}(1_{\mathsf{C}}, 1_{\mathsf{C}}). \tag{4.1}$$

When $\mathsf{C}$ is a monoidal $n$-category, we conceive a one-object $(n+1)$-category $B\mathsf{C}$ such that

$$\Omega B\mathsf{C} = \mathsf{C}. \tag{4.2}$$

Remarkably, if $\mathsf{C}$ is further symmetric, $B\mathsf{C}$ has a canonical symmetric monoidal structure. In this particular case, we can further define iterated $B^n\mathsf{C}$. The two operations $\Omega$ and $B$ are apparently borrowed from *looping* and *delooping* in algebraic topology. The reason why they are adopted will be clear shortly.

**Space and $n$-groupoid**

We can extract an $n$-category $\pi_{\leq n}X$ from each topological space $X$. Objects thereof are points in $X$, 1-morphisms are homotopies of objects (paths), 2-morphisms are homotopies of 1-morphisms (based homotopies of paths), 3-morphisms are homotopies of 2-morphisms (based homotopies of based homotopies of paths), and so on inductively until that $n$-morphisms are the homotopy classes of homotopies of $(n-1)$-morphisms. Note that we can allow $n = \infty$ by canceling terminating the induction and deleting the eventual command of taking the homotopy class in the above construction.

The $n$-category $\pi_{\leq n}X$ has a special property that all of its morphisms are invertible up to equivalence. In general, an $n$-category with such a property is called an $n$-groupoid. Thus $\pi_{\leq n}X$ is called the fundamental $n$-groupoid of $X$. This $\pi_{\leq n}X$ knows a great deal about the topology of $X$. First, the equivalence classes of objects in $\pi_{\leq n}X$ exactly constitute $\pi_0 X$. Second, for any base point $x \in X$, the fusion monoid of the equivalence classes of objects

in $\Omega^q \pi_{\leq n} X$ is exactly the homotopy group $\pi_q(X, x)$ for $q \leq n$ and the trivial group for $q > n$, accordingly. Furthermore, the entire first $n$-th stages of the Postnikov tower of each path component of $X$ are encoded in $\pi_{\leq n} X$. In other words, the fundamental $n$-groupoid completely determines the homotopy $n$-type of the space. Namely, $\pi_{\leq n} X \simeq \pi_{\leq n} Y$ as long as there is an $(n+1)$-connected map between $X$ and $Y$.

It is a classical theorem that every groupoid is equivalent to the fundamental groupoid of some space. Since Quillen [75] and Grothendieck [76], it has also been generally accepted that every $n$-groupoid is equivalent to the fundamental $n$-groupoid of some space. Therefore, based on the homotopical property discussed above, we have the following equivalence.

**Philosophy 4.1** *An $n$-groupoid is equivalent to a homotopy $n$-type. That is, an $n$-groupoid is an $n$-aspherical space, and the equivalence between $n$-groupoids is the weak homotopy equivalence between $n$-aspherical spaces.*

One can interpret this equivalence as saying that $n$-categories are non-invertible generalizations of homotopy $n$-types. Let us look at the particular case of $n = \infty$. Then $\pi_{\leq \infty} X$ encodes the entire Postnikov tower of $X$ and completely determines the homotopy type of $X$. The above equivalence then suggests that we can model $\infty$-groupoids by spaces.

**Philosophy 4.2** *An $\infty$-groupoid is equivalent to a homotopy type. That is, an $\infty$-groupoid is a space, and the equivalence between $\infty$-groupoids is the weak homotopy equivalence between spaces.*

An invertible monoidal $(n-1)$-groupoid is called an $n$-group. An $\infty$-group is equivalent to a loop space. $\Omega$ and $B$ establish a one-to-one correspondence between $n$-groups and one-object $(n+1)$-groupoids.

We also unify the symbols for spaces and higher-groupoids. We shall write $\pi_{\leq \infty} X$ just as $X$ and abandon the now tautological symbol $\pi_{\leq \infty} X$. We shall also write the homotopy $n$-type of $X$, incarnated by the $n$-th Postnikov truncation of $X$, just as $\pi_{\leq n} X$. It is reasonable to regard the higher-category theory as the non-invertible generalization of algebraic topology, in the sense that any correct model for higher-categories should produce philosophies 4.1 and 4.2 as theorems.

### $(n, r)$-**category**

We now learned that the invertibility of morphisms is a distinguished property for higher-categories. Thus people introduced the notion of $(n+r, n)$-categories to specify the information about invertibilities. From one perspective, $(n+r, n)$-category is just an $(n+r)$-category whose $p$-morphisms are invertible up to equivalence for all $p > n$. For example, an $(r, 0)$-category just means an $r$-groupoid and an $(n, n)$-category just means an $n$-category. For $n_1 \leq n_2 \leq n$, an $(n, n_1)$-category is also an $(n, n_2)$-category.

From another perspective, an $(n+r, n)$-category is an $n$-category enriched over $r$-groupoids. Namely, in the previous definition sketch of $n$-categories, we now choose to start the induction from $r$-groupoids instead of the mere sets. These $r$-groupoids are directly modeled by topological spaces according to philosophies 4.1 (and 4.2). This second perspective has bonus advantages in some aspects and has thus drawn vast attention. In particular, researches on $(\infty, n)$-categories are thriving.

## 4.2 Fully-extended TQFT

A general $n$-dim fully-extended TQFT is formulated as a symmetric monoidal functor between two symmetric monoidal $(\infty, n)$-categories, which axiomatizes the "results" of the path integral. The domain is a bordism $(\infty, n)$-category; the codomain is a fully-dualizable $(\infty, n)$-category. The specific physics context determines the choice of domains and codomains.

### 4.2.1 Bordism domain $(\infty, n)$-category

An $n$-tangential structure is a map $X \xrightarrow{\Gamma} BO(n)$. The simplest examples include

$$n\text{-framing:} \qquad \{*\} \xrightarrow{fr} BO(n) \,, \tag{4.3a}$$

$$n\text{-orientation:} \qquad \text{canonical } BSO(n) \xrightarrow{SO} BO(n) \,, \tag{4.3b}$$

$$n\text{-spin structure:} \qquad \text{canonical } BSpin(n) \xrightarrow{Spin} BO(n) \,. \tag{4.3c}$$

For an $n$-tangential structure $\Gamma$ and for $q \leq n$, a $q$-dim $\Gamma$ manifold means a $q$-dim manifold $M$ equipped with a map $M \to X$, such that the composition $M \to X \xrightarrow{\Gamma} BO(n)$ classifies the $n$-stabilized tangent bundle of $M$ (i.e. $TM \oplus \mathbb{R}^{n-q}$).

Given an $n$-tangential structure $\Gamma$, Lurie [64, Sec. 2.2] conceives a symmetric monoidal $(\infty, n)$-category $\mathsf{Bord}_n^\Gamma$ as follows. In the non-invertible-morphism region of $q \leq n$, a $q$-morphism is a $q$-dim $\Gamma$ bordism, i.e., a $q$-dim $\Gamma$ manifold with corners. In the invertible-morphism region, the homomorphism $\infty$-groupoids between $n$-morphisms are the spaces of boundary-fixed diffeomorphisms along the philosophy 4.2. The symmetric monoidal structure on $\mathsf{Bord}_n^\Gamma$ is prescribed by the disjoint union. Such $\mathsf{Bord}_n^\Gamma$'s give the domains of a fully-extended TQFT. However, given that the codomains we will be considering are essentially $n$-categories, the spaces of diffeomorphisms will not contribute to our results.

We want to equip all manifolds with a map to the target space $Y$. The simplest way to implement this is to adopt a special $n$-tangential structure $e_Y \times \Gamma : Y \times X \to BO(n)$, the product of the collapse map $Y \xrightarrow{e_Y} \{*\}$ and another $n$-tangential structure $X \xrightarrow{\Gamma} BO(n)$, such as those listed in Eq. (4.3). In this case, we shall particularly call $e_Y \times \Gamma$ manifolds $Y$-enriched $\Gamma$ manifolds, and particularly write

$$\mathsf{Bord}_n^\Gamma(Y) \equiv \mathsf{Bord}_n^{e_Y \times \Gamma} \,. \tag{4.4}$$

Such $\mathsf{Bord}_n^\Gamma(Y)$'s give the domains for $Y$-enriched fully-extended TQFTs. This notion of enriched TQFT is tightly related to symmetric TQFT (or say equivariant TQFT). Due to the dimensional reason, a map from a manifold of dimension $\leq n$ to $Y$ factors into $Y$'s homotopy $n$-type, $\pi_{\leq n} Y$. That is, we have the following equivalence,

$$\mathsf{Bord}_n^\Gamma(Y) \simeq \mathsf{Bord}_n^\Gamma(\pi_{\leq n} Y) \,. \tag{4.5}$$

If $Y$ is path connected, an $n$-dim $Y$-enriched TQFT is exactly an $n$-dim TQFT that acquires an action by the higher-group $\Omega \pi_{\leq n} Y$, i.e. an $\Omega \pi_{\leq n} Y$-symmetric TQFT. If $Y$ has multiple path components, we obtain a TQFT consisting of a $\pi_0 Y$ worth of universes, each of which is a higher-group-symmetric TQFT.

Physically, a priori, any TQFT suffers from a framing anomaly and requires a framing dependence. Therefore, $\mathsf{Bord}_n^{fr}(Y)$ would become a suitable choice of the domain for all TQFTs. A universal $Y$-enriched fully-extended TQFT with framing is a symmetric monoidal functor

$$\mathcal{Z} : \mathsf{Bord}_n^{fr}(Y) \to \mathsf{D}_n \, , \tag{4.6}$$

where the target category is a fully-dualizable symmetric monoidal $(\infty, n)$-category $\mathsf{D}_n$ as we shall see later. The maximal $\infty$-groupoid inside $\mathsf{D}_n$ acquires a natural homotopy $O(n)$-action according to the cobordism hypothesis [64, Theorem 2.4.6 and Corollary 2.4.10]. This homotopy $O(n)$-action can be lifted to a homotopy $\Omega X$-action by an $n$-tangential structure $X \xrightarrow{\Gamma} BO(n)$. If this $\Omega X$-action is (canonically) trivializable, the a priori framing anomaly turns out to be merely a $\Gamma$ anomaly, and then the TQFT $\mathcal{Z}$ does not really depend on an $n$-framing but an $n$-tangential structure $\Gamma$ instead. Consequently, in such a situation, we can (canonically) extend the domain of $\mathcal{Z}$ to any $Y$-enriched $\Gamma$ manifolds so that

$$\mathcal{Z}^\Gamma : \mathsf{Bord}_n^\Gamma(Y) \to \mathsf{D}_n \, . \tag{4.7}$$

Along this philosophy of treatment, what manifolds a TQFT can inhabit are determined by the property of its codomain[8].

### 4.2.2 Physical codomain $(\infty, n)$-category

Let $\mathsf{D}_n$ denote a fully-dualizable symmetric monoidal $(\infty, n)$-category that we want to use as the codomain for TQFTs (see Sec. 2.3 of Ref. [64] for the definition of full dualizability). For a physical TQFT from domain $\mathsf{Bord}_n^\Gamma(Y)$, we want to assign a complex number to each closed $n$-dim $Y$-enriched $\Gamma$ manifold as its partition function, and assign a vector space to each closed $(n-1)$-dim $Y$-enriched $\Gamma$ manifolds as its state space. In other words, we want $\Omega^{n-1}\mathsf{D}_n$ to be an $(\infty, 1)$-completion of a symmetry monoidal category of proper vector spaces. There are two inequivalent natural $(\infty, 1)$-completions of a category of vector spaces.

- Only identity higher morphisms are added. $\mathrm{Hom}(-, -)$ has the discrete topology.

- Iterated isotopies are added. $\mathrm{Hom}(-, -)$ has the subspace topology from some $\mathbb{C}^m$.

The second completion is appropriate for classifying gapped phases because it identifies TQFT deformation classes[9]. However, the first completion is appropriate for our purpose because we want to speak of the partition function of each individual TQFT. Thus in this paper, we regard the first completion as the canonical $(\infty, 1)$-completion and abuse the same symbol of the category to denote also its canonical $(\infty, 1)$-completion.

---

[8]This treatment may be phrased as codomain-dominated. There is also a domain-dominated treatment, where we take a universal codomain $\mathsf{D}_n$ (for each $n$) and ask $\Gamma$ to vary. Then the logic will be reversed, i.e., the choice of $\Gamma = SO$ (resp. $\Gamma = Spin$) implies that bosonic (resp. fermionic) state spaces will be picked up in the universal codomain. Since a universal codomain is challenging to construct, we take the codomain-dominated approach.

[9]It captures deformation classes of not only orthodox TQFTs but also half-geometric-half-topological QFTs. These unorthodox "TQFTs" depend on geometric data of background fields $\sigma : M \mapsto Y$ but are invariant under spacetime diffeomorphisms. This is exactly what is needed for classifying gapped phases.

The choice of $\Omega^{n-1}\mathsf{D}_n$ is determined by what state spaces we want. In the bosonic case, we consider $\Omega^{n-1}\mathsf{D}_n \simeq \mathsf{Vect}^{fd}$ whose objects are finite-dim $\mathbb{C}$-linear spaces and morphisms are $\mathbb{C}$-linear maps. The tensor product gives rise to a monoidal structure on $\mathsf{Vect}^{fd}$. A swap map comes from the index exchange,

$$x \otimes y \;\mapsto\; y \otimes x \,, \tag{4.8}$$

which makes $\mathsf{Vect}^{fd}$ a symmetric monoidal category. All objects have duals due to the finite-dim condition. It also has finite biproducts given by the direct sum, is semisimple with respect to it, and has a unique class of simple objects. It is further $\mathbb{C}$-linear. All these structures make $\mathsf{Vect}^{fd}$ a symmetric fusion category.

As for the fermionic case, the state space $V$ has a $\mathbb{C}$-linear involution $\varepsilon$ called fermionic parity, which makes the state space $\mathbb{Z}_2$-graded. Such a pair $(V,\varepsilon)$ is called a super vector space. The $(+1)$-eigenspace of $\varepsilon$ is the bosonic sector and the $(-1)$-eigenspace is the fermionic sector. Let $\mathsf{sVect}^{fd}$ denote the category whose objects are finite-dim super $\mathbb{C}$-linear spaces and morphisms are $\mathbb{C}$-linear maps that commute with fermionic parities. Its monoidal structure also comes from the tensor product,

$$\big(X,\varepsilon_X\big) \otimes \big(Y,\varepsilon_Y\big) \simeq \big(X\otimes Y,\, \varepsilon_X \otimes \varepsilon_Y\big)\,. \tag{4.9}$$

$\mathsf{sVect}^{fd}$ is also $\mathbb{C}$-linear, has finite biproducts, and has duals for all objects, making it a fusion category. As fusion categories, $\mathsf{sVect}^{fd}$ is equivalent to $\mathsf{Rep}(\mathbb{Z}_2)$, the representation category of $\mathbb{Z}_2$. It is the swap map that distinguishes them as inequivalent symmetric fusion categories. The swap map in $\mathsf{sVect}^{fd}$ encodes the fermionic statistics following the Koszul sign rule. Namely, for fermionic parity eigenstates $x \in X$ and $y \in Y$, the swap map sends

$$x \otimes y \;\mapsto\; (-)^{|x||y|}\, y \otimes x\,, \tag{4.10}$$

where $|\bullet|=0$ if $\bullet$ is bosonic and $|\bullet|=1$ if $\bullet$ is fermionic. We take $\Omega^{n-1}\mathsf{D}_n \simeq \mathsf{sVect}^{fd}$ for the fermionic case.

The final step is to determine the entire $\mathsf{D}_n$. Since we do not have any requirement on higher objects than particles, our guiding principle is that no superfluous data on lower-dim manifolds, except when necessary, should be introduced. Gaiotto and Johnson-Freyd found an elegant solution in Ref. [68] (see also Ref. [77]). They noticed the Karoubi-completeness among various other features and clarified that the minimal necessary complexity is exactly the $n$-categorical generalization of being Karoubi-complete. Roughly speaking, being Karoubi-complete means that every idempotent comes from a splitting, among all $p$-morphisms. Given a Karoubi-complete monoidal $n$-category $\mathsf{C}$, they defined its "stable suspension" $\Sigma\mathsf{C}$ as the Karoubi completion of its delooping $B\mathsf{C}$. Symbolically,

$$\Sigma\mathsf{C} \equiv \mathrm{Kar}(B\mathsf{C})\,. \tag{4.11}$$

When the above $n$-category $\mathsf{C}$ is further symmetric, $\Sigma\mathsf{C}$ turns out to be a Karoubi-complete symmetric monoidal $(n+1)$-category [68, Theorem 4.1.1]. Given that both $\mathsf{Vect}^{fd}$ and

$\mathsf{sVect}^{fd}$ are Karoubi-complete, the codomain $\mathsf{D}_n$ we are looking for can be given by

$$(\text{bosonic}) \quad \Sigma^{n-1}\mathsf{Vect}^{fd}, \tag{4.12a}$$

$$(\text{fermionic}) \quad \Sigma^{n-1}\mathsf{sVect}^{fd}. \tag{4.12b}$$

We shall flexibly regard them as either $n$-categories or $(\infty, n)$-categories (with identity higher morphisms) according to the context. Gaiotto and Johnson-Freyd also conjecture [68, Conjecture 1.4.6] that the canonical homotopy $SO(n)$-action [resp. $Spin(n)$-action] on the maximal $\infty$-groupoid inside $\Sigma^{n-1}\mathsf{Vect}^{fd}$ (resp. $\Sigma^{n-1}\mathsf{sVect}^{fd}$) is canonically trivializable. If these conjectured properties are true, according to the discussion around Eq. (4.7), bosonic (resp. fermionic) topological functionals indeed inhabit any closed oriented (resp. spin) manifold.

### 4.2.3 Summary of the formulation

We now summarize all the ingredients found above to present accurate formulations of the relevant TQFTs to clarify the connotation of Ansatz 3.2. A priori, any TQFT suffers from a framing anomaly and requires a framing dependence.

**Definition 4.3** *An $n$-dim bosonic (resp. fermionic) $Y$-enriched fully-extended TQFT is a symmetric monoidal functor between symmetric monoidal $(\infty, n)$-categories,*

$$\mathcal{B} : \mathsf{Bord}_n^{fr}(Y) \to \Sigma^{n-1}\mathsf{Vect}^{fd}, \qquad \left[resp. \quad \mathcal{F} : \mathsf{Bord}_n^{fr}(Y) \to \Sigma^{n-1}\mathsf{sVect}^{fd}\right]. \tag{4.13}$$

It is reasonable to anticipate that having bosonic (resp. fermionic) state spaces logically implies inhabiting any oriented (resp. spin) spacetime, i.e., the framing anomaly should merely be an orientation (resp. spin) anomaly. This anticipation is realized by the following conjectured property of $\Sigma^{n-1}\mathsf{Vect}^{fd}$ and $\Sigma^{n-1}\mathsf{sVect}^{fd}$ [68]:

**Conjecture 4.4** *The canonical homotopy $SO(n)$-action [resp. $Spin(n)$-action] on the maximal $\infty$-groupoid inside $\Sigma^{n-1}\mathsf{Vect}^{fd}$ (resp. $\Sigma^{n-1}\mathsf{sVect}^{fd}$) is canonically trivializable.*

By the cobordism hypothesis combined with this conjecture, every $\mathcal{B}$ (resp. $\mathcal{F}$) has no genuine framing dependence but just an orientation (resp. spin) dependence, so every $\mathcal{B}$ or $\mathcal{F}$ can be canonically extended to

$$\mathcal{B}^{SO} : \mathsf{Bord}_n^{SO}(Y) \to \Sigma^{n-1}\mathsf{Vect}^{fd}, \tag{4.14a}$$

$$\mathcal{F}^{Spin} : \mathsf{Bord}_n^{Spin}(Y) \to \Sigma^{n-1}\mathsf{sVect}^{fd}. \tag{4.14b}$$

Therefore, we will talk about the partition functions of $\mathcal{B}$ (resp. $\mathcal{F}$) on any closed oriented (resp. spin) $n$-manifold $M$, which really mean those of $\mathcal{B}^{SO}$ (resp. $\mathcal{F}^{Spin}$).

When $Y$ is path-connected, the evaluation of $\mathcal{B}$ (resp. $\mathcal{F}$) on a point gives a linear $n$-group action on objects in $\Sigma^{n-1}\mathsf{Vect}^{fd}$ (resp. $\Sigma^{n-1}\mathsf{sVect}^{fd}$) imposed by the $n$-group $\Omega\pi_{\leq n}Y$. Namely, an $n$-dim $Y$-enriched TQFT is the same as an $n$-dim $\Omega\pi_{\leq n}Y$-symmetric TQFT. When $Y$ has multiple path components, we have a $\pi_0(Y)$ worth of universes of higher-group symmetric TQFTs. The cobordism hypothesis [64] asserts that such a characterization by evaluation on a point is always faithful and complete. In general, $\mathsf{Bord}_n^{fr}(Y)$ is the free fully-dualizable symmetric monoidal $(\infty, n)$-category generated by $\infty$-groupoid $Y$.

**Definition 4.5** *A $n$-representation (resp. super $n$-representation) of space $Y$ is a functor between $(\infty, n)$-categories,*

$$\underline{\mathcal{B}} : Y \to \Sigma^{n-1}\mathsf{Vect}^{fd}, \qquad \left(resp. \quad \underline{\mathcal{F}} : Y \to \Sigma^{n-1}\mathsf{sVect}^{fd}\right). \tag{4.15}$$

Note that since $\Sigma^{n-1}\mathsf{Vect}^{fd}$ (resp. $\Sigma^{n-1}\mathsf{sVect}^{fd}$) is essentially an $n$-category, $\underline{\mathcal{B}}$ (resp. $\underline{\mathcal{F}}$) actually factors through the homotopy $n$-type of $Y$. Namely, $\underline{\mathcal{B}}$ (resp. $\underline{\mathcal{F}}$) is in essence a functor between $n$-categories, from $\pi_{\leq n}Y$ to $\Sigma^{n-1}\mathsf{Vect}^{fd}$ (resp. $\Sigma^{n-1}\mathsf{sVect}^{fd}$). The cobordism hypothesis specialized for our purpose asserts the following equivalence (see [64, Theorem 2.4.18]):

**Proposition 4.6** *Via point evaluation, an $n$-dim bosonic (resp. fermionic) $Y$-enriched fully-extended TQFT (Definition 4.3) is equivalent to a $n$-representation (resp. super $n$-representation) of $Y$ (Definition 4.5).*

Higher-representations at lower dimensions in the context of generalized symmetries are extensively discussed by, e.g., Refs. [57, 59].

As a preliminary example of topological functionals from these TQFTs, let us consider 1D topological functionals to path-connected 1-finite $Y$. Namely, $\pi_1 Y$ is finite. We start with the bosonic case. A 1-representation $\underline{\mathcal{B}}$ of $Y$ is just a representation $R$ of $\pi_1 Y$. An $S^1$ bosonic topological functional $\mathcal{B}(g)$ for $g \in [S^1, Y]$ is a $\mathbb{C}$-valued function on

$$[S^1, Y] \simeq [S^1, B\pi_1 Y] \simeq \{\text{conjugacy classes of } \pi_1 Y\}. \tag{4.16}$$

Note that $[S^1, Y] \not\simeq \pi_1 Y$ as long as $\pi_1 Y$ is not commutative[10]. Therefore, an $S^1$ topological functional is equivalent to a class function of $\pi_1 Y$, such as a character. Note that $g \in [S^1, Y]$ prescribes a $\pi_1 Y$ gauge field on $S^1$ whose holonomy conjugacy class is $g$. Therefore, the partition function is indeed given by a character, i.e.,

$$\mathcal{B}(g) = \mathrm{tr}_R(g). \tag{4.17}$$

One can generalize this relation to $T^n$ bosonic topological functionals and define the notion of $n$-characters[11]. The traces of representations amount to all characters, and the space of characters is $\mathbb{N}$-linearly spanned by simple characters. The characters from 1-dim representations factor through $H_1(Y; \mathbb{Z})$, the Abelianization of $\pi_1 Y$, and thus can be constructed via the universal invertible form (2.7). The characters from higher-dim representations, especially the irreducible ones, go beyond Eq. (2.7).

For the fermionic case, a super 1-representation $\underline{\mathcal{F}}$ of $Y$ is just a super representation of $\pi_1(Y)$, which is further a pair of ordinary representations $(B, F)$ on the bosonic and the

---

[10]This discrepancy between $[S^1, Y]$ and $\pi_1 Y$ comes from the difference between free homotopy and based homotopy. As we shall see in Sec. 5.2.1, this difference accounts for a large class of non-invertible fusion rules in solitonic symmetry.

[11]Ideas of $n$-characters originate from Ref. [78]. 2-characters are defined in Ref. [79] and are developed in, e.g., Refs. [80, 81]. Unlike ordinary characters, the input of an $n$-character is a set of $n$ mutually commutative elements in $G$, and the $n$-character is invariant under simultaneous conjugations. One can immediately recognize that this input describes the isomorphism classes of $G$-bundles on $T^n$, i.e., $[T^n, BG]$. It is then natural to connect these $n$-characters with $T^n$ bosonic topological functionals to $BG$.

fermionic sectors, respectively. There are two spin structures on $S^1$, 0 and 1, corresponding to the spin bordism group $\Omega_1^{Spin} = \mathbb{Z}_2$. Then a similar analysis to the bosonic case shows that the $S^1$ fermionic topological functional for $(g, \varepsilon) \in [S^1, Y] \times \Omega_1^{Spin}$ is given by [recall that $g$ represents a conjugacy class in $\pi_1 Y$]

$$
\begin{aligned}
\mathcal{F}(g, 0) &= \text{tr}_B(g) + \text{tr}_F(g)\,, \\
\mathcal{F}(g, 1) &= \text{tr}_B(g) - \text{tr}_F(g)\,.
\end{aligned}
\tag{4.18}
$$

We thus learned that $S^1$ fermionic topological functionals are virtual characters of $\pi_1 Y$. They are $\mathbb{Z}$-linearly spanned by simple characters rather than $\mathbb{N}$-linearly. The virtual characters from 1-dim representations factor through the super integral homology $\mathbb{SZ}_1(Y)$ (see Sec. 4.3.2) and thus can also be constructed via the universal invertible form (2.7). Tremendous examples of higher-dim topological functionals will be discussed in Sec. 5.

## 4.3 Cohomology with TQFT coefficients

As we mentioned in Sec. 2.2.1, a crucial feature of solitonic symmetry is its independence of system details such as the action and the ambient spacetime. It is just the algebra of topological functionals and is determined by the target space $Y$ only. Formally, it gives homotopy-invariant contravariant functors on the topological space $Y$. Thus the algebraic structure of solitonic symmetry can be interpreted as a cohomology theory on $Y$, in a vastly generalized sense.

### 4.3.1 Non-invertible: (Super) solitonic cohomology

Given two TQFTs $\mathcal{Z}_1$ and $\mathcal{Z}_2$, $\mathcal{Z}_1(-) \otimes \mathcal{Z}_2(-)$ also defines a TQFT denoted by $\mathcal{Z}_1 \otimes \mathcal{Z}_2$. We can transfer the fusion of two topological functionals to the fusion of the two TQFTs beneath. To see the tensor product between TQFTs, instead of looking at each individual TQFT, we should consider the functor $(\infty, n)$-category containing all $n$-representations[12]. It is actually an $n$-category because of the $n$-category nature of the codomain.

**Definition 4.7** *The $n$-th solitonic cohomology [resp. super solitonic cohomology] of an $n$-finite space $Y$ is a symmetric multi-fusion $n$-category,*

$$
\text{Rep}^n(Y) \equiv \text{Fun}\left(Y, \Sigma^{n-1}\text{Vect}^{fd}\right), \quad \left[resp.\ \text{sRep}^n(Y) \equiv \text{Fun}\left(Y, \Sigma^{n-1}\text{sVect}^{fd}\right)\right]. \tag{4.20}
$$

They are symmetric fusion $n$-categories when $Y$ is further path-connected. In this case, they can be understood as higher-representation higher-categories of the $\infty$-group $\Omega Y$. In particular, the $n$-category nature of the codomains suggests the following.

- When $Y$ is path-connected, $\text{Rep}^1(Y)$ [resp. $\text{sRep}^1(Y)$] is equivalent to the representation (resp. super-representation) category of the group $\pi_1 Y$.

---

[12]One may first come up with

$$
\text{Fun}^\otimes\left(\text{Bord}_n^{fr}(Y), \Sigma^{n-1}\text{Vect}^{fd}\right) \quad \text{and} \quad \text{Fun}^\otimes\left(\text{Bord}_n^{fr}(Y), \Sigma^{n-1}\text{sVect}^{fd}\right). \tag{4.19}
$$

According to the cobordism hypothesis, they are the maximal $\infty$-groupoids (in essence $n$-groupoids) inside $\text{Rep}^n(Y)$ and $\text{sRep}^n(Y)$, respectively. They have too few morphisms to support the rich structures we shall discuss shortly.

- When $Y$ is path-connected, $\mathsf{Rep}^n(Y)$ [resp. $\mathsf{sRep}^n(Y)$] is equivalent to the representation (resp. super-representation) $n$-category of the $n$-group $\Omega\pi_{\leq n}Y$.

The 1-dim case reproduces the discussion on 1-dim topological functionals at the end of Sec. 4.1. The higher-group representation higher-category has caught vast attention in recent literature on generalized symmetry, see, e.g., Refs. [52–55, 57, 59]. When $Y$ is not path-connected, the solitonic cohomologies are just the Cartesian product of the solitonic cohomologies of each path component.

The fusion monoid of bosonic (resp. fermionic) topological functionals can be equated with the fusion monoid of the equivalence classes of objects in $\mathsf{Rep}^n(Y)$ [resp. $\mathsf{sRep}^n(Y)$], which gives what we have been pursuing. However, we should not quickly throw away the far richer structures contained in $\mathsf{Rep}^n(Y)$ and $\mathsf{sRep}^n(Y)$ than their mere fusion monoids. To see their significance, let us analyze the physical meaning of morphisms in $\mathsf{Rep}^n(Y)$ and $\mathsf{sRep}^n(Y)$. Here 1-morphisms are natural transformations between $n$-representations. We can readily note that the natural endo-transformations of the trivial $n$-representation are simply functors from $Y$ to $\Sigma^{n-2}\mathsf{Vect}^{fd}$ and $\Sigma^{n-2}\mathsf{sVect}^{fd}$, respectively. Namely, we arrive at the following relations between different $n$.

**Proposition 4.8** $\Omega\mathsf{Rep}^n(Y) \simeq \mathsf{Rep}^{n-1}(Y)$ *and* $\Omega\mathsf{sRep}^n(Y) \simeq \mathsf{sRep}^{n-1}(Y)$.

This result hints at the physical meaning of other morphisms in $\mathsf{Rep}^n(Y)$ and $\mathsf{sRep}^n(Y)$:

- 1-morphisms are $(n-1)$-dim topological interfaces between $n$-dim TQFTs.

- $(p+1)$-morphisms are $(n-p-1)$-dim topological interfaces between $(n-p)$-dim topological interfaces.

Topological functionals, which are auxiliary-TQFT partition functions defined on proper closed manifolds, cannot capture these richer data, which concern auxiliary TQFTs themselves and inhabit networks made by non-closed manifolds via connections and junctions. In recent literature on generalized symmetry, people have recognized the significance of such networks of non-closed topological operators as a formulation of background gauge fields. It starts to become customary to recognize the algebraic structure of such networks as the total generalized symmetry of a theory, given that it contains literally all information about a symmetry, not just the fusion rule. We have no reason not to follow this custom.

**Proposition 4.9** *Consider a $d$-dim bosonic (resp. fermionic) theory defined by a path integral with $d$-finite target space $Y$.*

- *The total solitonic symmetry is described by $\mathsf{Rep}^d(Y)$ [resp. $\mathsf{sRep}^d(Y)$], a symmetric fusion $d$-category.*

- *For $-1 \leq p \leq d-1$, the $(\geq p)$-form solitonic symmetry is described by $\mathsf{Rep}^{d-p-1}(Y)$ [resp. $\mathsf{sRep}^{d-p-1}(Y)$], a symmetric fusion $(d-p-1)$-category.*

- *For $-1 \leq p \leq d-1$, the $p$-form solitonic symmetry is described by the fusion monoid of $\mathsf{Rep}^{d-p-1}(Y)$ [resp. $\mathsf{sRep}^{d-p-1}(Y)$], a commutative rig.*

Note that we also include (–1)-form symmetry in the total solitonic symmetry. The result here echoes recent progresses on categorical generalized symmetry.

We may understand $\mathsf{Rep}^\bullet(-)$ and $\mathsf{sRep}^\bullet(-)$ as non-invertible generalizations of cohomology theories. We may view the collections,

$$\left\{\Sigma^{n-1}\mathsf{Vect}^{fd}\right\}_{n\in\mathbb{N}} \text{ and } \left\{\Sigma^{n-1}\mathsf{sVect}^{fd}\right\}_{n\in\mathbb{N}}, \tag{4.21}$$

as non-invertible generalizations of spectra. We shall shortly see in Sec. 4.3.2 which orthodox spectra they generalize. The coefficients of these non-invertible cohomologies are non-enriched fully-extended TQFTs:

$$\begin{aligned}
\mathsf{Rep}^\bullet &\equiv \mathsf{Rep}^\bullet(\{*\}) \simeq \Sigma^{\bullet-1}\mathsf{Vect}^{fd}, \\
\mathsf{sRep}^\bullet &\equiv \mathsf{sRep}^\bullet(\{*\}) \simeq \Sigma^{\bullet-1}\mathsf{sVect}^{fd}.
\end{aligned} \tag{4.22}$$

$\mathsf{Rep}^\bullet(-)$ and $\mathsf{sRep}^\bullet(-)$ are indeed homotopy-invariant contravariant functors from topological spaces because a map $Y \to Z$ induces pullbacks $\mathsf{Rep}^\bullet(Z) \to \mathsf{Rep}^\bullet(Y)$ and $\mathsf{sRep}^\bullet(Z) \to \mathsf{sRep}^\bullet(Y)$ simply by their definitions.

When $Y$ is not $n$-finite, Def. 4.7 contains too much superfluous data to capture the authentic algebraic structure of solitonic symmetry. For example, recall the discussion about 1-dim topological functionals at the end of Sec. 4.2.3. Recall that $\mathsf{Rep}^1(Y)$ and $\mathsf{sRep}^1(Y)$ are simply representation categories of $\pi_1 Y$. If we allowed $\pi_1 Y$ to be infinite like $\mathbb{Z}$ or $SL(2,\mathbb{Z})$, non-semisimple representations would appear since the Maschke theorem does not apply. They are not unitarizable and produce superfluous duplicated topological functionals. There are also semisimple non-unitarizable representations whose $S^1$ partition functions are not bounded. It is natural to expect that only unitarizable representations capture the solitonic symmetry. Nevertheless, for convenience, we still formally use $\mathsf{Rep}^\bullet(Y)$ or $\mathsf{sRep}^\bullet(Y)$ to indicate the algebraic structure of solitonic symmetry even when $Y$ is not $\bullet$-finite.

We do not attempt to generalize the above unitarizability condition to higher-dim general cases in this paper. Instead, as we discussed at the end of Sec. 3.3, we shall just focus on $n$-finite homotopy quotients $Z$ of $Y$ and discuss the solitonic subsymmetries that can be faithfully described by $\mathsf{Rep}^\bullet(Z)$ or $\mathsf{sRep}^\bullet(Z)$. The collection of them for all different choices of $Z$, together with natural functors between them, form a diagram in the $(n+1)$-category of symmetric multi-fusion $n$-categories. We expect that the colimit of this diagram exists and believe that this colimit gives an almost complete approximation of the continuous solitonic symmetry for $Y$. Furthermore, we expect that the continuous solitonic symmetry may be constructed as a generalized Cauchy completion of this colimit, provided we can cook up a well-behaved notion of topologized/uniformized monoidal $n$-categories[13].

### 4.3.2 Invertible: (Super) unitary cohomology

As the first application, let us find out the invertible subsymmetry of the generically non-invertible solitonic symmetry. Invertible topological functionals are the partition functions

---

[13] The prototype of our anticipation is just the Cauchy completion of $\mathbb{Q}/\mathbb{Z}$ to $U(1)$. It is possible to prescribe a topology on $\mathbb{Q}/\mathbb{Z}$ through the colimit construction $\mathbb{Q}/\mathbb{Z} \simeq \bigcup_{n\in\mathbb{N}}\mathbb{Z}_n$. Topological groups are naturally uniformizable; thus, we can take the Cauchy completion of $\mathbb{Q}/\mathbb{Z}$ to obtain $U(1)$.

of invertible fully-extended TQFTs. A fully-extended TQFT $\mathcal{Z} : \mathsf{Bord}_n^\Gamma \to \mathsf{D}_n$ is said invertible if we can find another fully-extended TQFT $\mathcal{Z}^{-1} : \mathsf{Bord}_n^\Gamma \to \mathsf{D}_n$ such that $\mathcal{Z} \otimes \mathcal{Z}^{-1} \simeq 1$, the trivial TQFT. Therefore, $\mathcal{Z}$ must factor through the following commutative diagram (see the discussion around Sec. 6.2 of Ref. [67]):

$$
\begin{array}{ccc}
\mathsf{Bord}_n^\Gamma & \xrightarrow{\ \mathcal{Z}\ } & \mathsf{D}_n \\
\downarrow & & \uparrow \\
\|\mathsf{Bord}_n^\Gamma\| & \xrightarrow{\ \mathcal{Z}^\times\ } & \mathsf{D}_n^\times
\end{array}
\tag{4.23}
$$

Here, $\mathsf{D}_n^\times$ is the maximal Picard $\infty$-groupoid inside $\mathsf{D}_n$, where a Picard $\infty$-groupoid means an invertible symmetric monoidal $\infty$-groupoid (see Appendix A.4 of Ref. [67]). And $\|\mathsf{Bord}_n^\Gamma\|$ is the $\infty$-groupoid completion of $\mathsf{Bord}_n^\Gamma$, which turns out to have invertible objects only and becomes a Picard $\infty$-groupoid. Thus $\mathcal{Z}^\times$ is a symmetric monoidal functor between two Picard $\infty$-groupoids.

According to Philosophy 4.2, a Picard $\infty$-groupoid is an infinite loop space, i.e., the 0-space of a spectrum. Also, $\mathcal{Z}^\times$ is an infinite loop map and the fusion between $\mathcal{Z}_1^\times$ and $\mathcal{Z}_2^\times$ is induced by loop concatenation. Therefore, classifying invertible TQFTs reduces to classifying infinite loop maps between two infinite loop spaces, which further reduces to classifying spectrum maps between two spectra. Therefore, the treatment of invertible TQFTs belongs to the realm of stable homotopy theory.

The Galatius-Madsen-Tillmann-Weiss theorem [82], a theorem derived out of the cobordism hypothesis [64], asserts that $\|\mathsf{Bord}_n^\Gamma\|$ is weakly homotopy equivalent to the 0-space of the Madsen-Tillmann spectrum $\Sigma^n \mathbb{M}\mathbb{T}\Gamma$ (see also [67, Theorem 6.67]). In the case of our interest, $\|\mathsf{Bord}_n^{fr}(Y)\|$ is independent of $n$ and is the free infinite loop space generated by $Y_+ \equiv Y \sqcup \{*\}$, i.e. $Y$ with an extra base point. Namely,

$$
\|\mathsf{Bord}_n^{fr}(Y)\| \simeq \operatornamewithlimits{colim}_{q \to \infty} \Omega^q \Sigma^q Y_+ \,.
\tag{4.24}
$$

It is the 0-space of the suspension spectrum of $Y_+$. The codomain is more complicated due to the complexity of the higher-categorical Karoubi completion. We thus leave the full analysis of the maximal invertible subsymmetry to future works. Here we just focus on a sufficiently interesting part we can obtain immediately by noticing

$$
\Omega^{n-1} \left( \Sigma^{n-1} \mathsf{D} \right)^\times \simeq \left( \Omega^{n-1} \Sigma^{n-1} \mathsf{D} \right)^\times \simeq \mathsf{D}^\times \,.
\tag{4.25}
$$

In general, the spectrum $\{ \left( \Sigma^{n-1} \mathsf{D} \right)^\times \}_{n \in \mathbb{N}}$ may have lousy connectivity, i.e., the Karoubi-completion procedure may add new invertible objects and make many $\left( \Sigma^{n-1} \mathsf{D} \right)^\times$ not path-connected. However, here we shall neglect such contributions from the Karoubi-completion procedure. In other words, we shall consider spectrum

$$
\{ B^{n-1} \mathsf{D}^\times \}_{n \in \mathbb{N}} \,,
\tag{4.26}
$$

which is the $(-1)$-connective cover of spectrum $\{ \left( \Sigma^{n-1} \mathsf{D} \right)^\times \}_{n \in \mathbb{N}}$. We thus obtain at least part of the maximal invertible subsymmetry[14].

---

[14] We cannot help speculating a relationship between $\{ \left( \Sigma^{n-1} \mathsf{Vect}^{fd} \right)^\times \}_{n \in \mathbb{N}}$ [resp. $\{ \left( \Sigma^{n-1} \mathsf{sVect}^{fd} \right)^\times \}_{n \in \mathbb{N}}$] and the $\mathbb{C}^\times$-dual of $\mathbb{M}SO$ (resp. $\mathbb{M}Spin$). We leave the verification of our speculations to future works.

Let us start with the bosonic case. Only the 1-dim vector space is invertible in $\mathsf{Vect}^{fd}$, and its invertible endomorphisms constitute the group $\mathbb{C}^\times$. Therefore, we have

$$B^{n-1}\left(\mathsf{Vect}^{fd}\right)^\times \simeq B^n\mathbb{C}^\times_\delta. \tag{4.27}$$

When ambiguity may arise, we use the subscript $\delta$ to indicate the discrete topology. These infinite loop spaces assemble to form the Eilenberg–Maclane spectrum $\mathbb{HC}^\times$. We then arrived at the following theorem:

**Proposition 4.10 (bosonic)** *The fusion monoid of solitonic cohomology* $\mathsf{Rep}^\bullet(Y)$ *contains a group* $(\mathbb{HC}^\times)^\bullet(Y) \simeq H^\bullet\left(Y; \mathbb{C}^\times\right)$.

We now turn to the more interesting fermionic case. $\mathsf{sVect}^{fd}$ has two classes of invertible objects, the bosonic and the fermionic 1-dim vector spaces. Their fusion monoid is $\mathbb{Z}_2$. The invertible endomorphisms of each class constitute $\mathbb{C}^\times$. The symmetric monoidal structure on $\mathsf{sVect}^{fd}$, especially the Koszul sign rule (4.10), positions the infinite loop space $\left(\mathsf{sVect}^{fd}\right)^\times$ into a Puppe sequence,

$$\cdots \longrightarrow B^n\mathbb{C}^\times_\delta \longrightarrow B^{n-1}\left(\mathsf{sVect}^{fd}\right)^\times \longrightarrow B^{n-1}\mathbb{Z}_2 \xrightarrow{\rho\circ\mathrm{Sq}^2} B^{n+1}\mathbb{C}^\times_\delta \longrightarrow \cdots, \tag{4.28}$$

classified by the stable cohomology operation $\rho \circ \mathrm{Sq}^2$.[15] Here $\mathrm{Sq}^2$ is the second Steenrod square, and $\rho$ is the change-of-coefficient for the canonical inclusion $\mathbb{Z}_2 \to \mathbb{C}^\times$. These infinite loop spaces assemble to form a spectrum and let us denote it as $\mathbb{SC}$. We have thus proved the following theorem:

**Proposition 4.11 (fermionic)** *The fusion monoid of super solitonic cohomology* $\mathsf{sRep}^\bullet(Y)$ *contains a group* $\mathbb{SC}^\bullet(Y)$, *where* $\mathbb{SC}$ *is defined via Eq.* (4.28).

For a $d$-finite target space $Y$, these theorems tell us that there are invertible $(d-n-1)$-form solitonic symmetry given by $H^n(Y; \mathbb{C}^\times)$ in a $d$-dim bosonic theory and $\mathbb{SC}^n(Y)$ in a $d$-dim fermionic theory. We can further ask what higher-group they make up, which requires us to consider the entire map spectrum $\mathrm{Map}(\mathbb{Y}_+, \mathbb{HC})$ or $\mathrm{Map}(\mathbb{Y}_+, \mathbb{SC})$, where $\mathbb{Y}_+$ denotes the suspension spectrum of $Y$, rather than merely its $\pi_0-$. We again leave such analysis to future works.

When we focus on $\bullet$-finite spaces only, many other cohomology theories can also provide the same results as above. For example, for $\bullet$-finite space $Y$ we have

$$H^\bullet(Y; \mathbb{C}^\times) \simeq H^\bullet\left(Y; U(1)\right), \tag{4.29a}$$

$$\mathbb{SC}^\bullet(Y) \simeq \mathbb{SU}^\bullet(Y), \tag{4.29b}$$

but they are different for general $Y$. Here $\mathbb{SU}$ is defined as the spectrum obtained by substituting $\mathbb{C}^\times$ with $U(1)$ in Eq. (4.28) and changing $\rho$ into the change-of-coefficient for $\mathbb{Z}_2 \to U(1)$. We shall call $\mathbb{HU}(1)$ the *unitary* spectrum and call $\mathbb{SU}$ the *super unitary* spectrum. We conjecture that these two spectra correctly capture invertible solitonic symmetry even when we remove the finiteness condition on $Y$.

---

[15]The direct determination of $\rho \circ \mathrm{Sq}^2$ requires to evaluate the $E_\infty$-structure of $\left(\mathsf{sVect}^{fd}\right)^\times$. The Koszul sign rule should prescribe an $E_\infty$-structure that leads to a nontrivial stable cohomology operation. We note however that $\rho \circ \mathrm{Sq}^2$ is the only existing nontrivial stable cohomology operation from $\mathbb{HZ}_2$ to $\Sigma^2\mathbb{HC}^\times$.

**Conjecture 4.12** *Consider a d-dim bosonic (resp. fermionic) theory defined by a path integral with target space $Y$.*

- *For $-1 \leq p \leq d-1$, the p-form solitonic symmetry contains the unitary cohomology [resp. super unitary cohomology] group $H^{d-p-1}(Y; U(1))$ [resp. $\mathbb{SU}^{d-p-1}(Y)$].*

*(Note: This is a theorem mentioned above when $Y$ is d-finite.)*

This conjecture is motivated by our discrete approximation discussed at the end of Sec. 3.3 and Sec. 4.3.1. They are also consistent with physicists' long experience with invertible $\theta$-angles and the universal invertible construction (2.7).

Before concluding this section, let us unpack these two cohomology theories. $\mathbb{H}U(1)$ is the U(1)-dual of $\mathbb{H}\mathbb{Z}$ and its universal coefficient theorem takes the naive form,

$$H^\bullet\big(-;U(1)\big) \simeq \mathrm{Hom}\big(H_\bullet(-,\mathbb{Z}), U(1)\big). \tag{4.30}$$

As for the fermionic case, the modified version of the Puppe sequence (4.28) leads to a long exact sequence[16],

$$\cdots \longrightarrow H^\bullet\big(-;U(1)\big) \longrightarrow \mathbb{SU}^\bullet(-) \longrightarrow H^{\bullet-1}\big(-;\mathbb{Z}_2\big) \overset{\rho\circ\mathrm{Sq}^2}{\longrightarrow} H^{\bullet+1}\big(-;U(1)\big) \longrightarrow \cdots . \tag{4.31}$$

$\mathbb{SU}$ is the U(1)-dual of $\mathbb{SZ}$, which we call the *super integral* spectrum. Namely,

$$\mathbb{SU}^\bullet(-) \simeq \mathrm{Hom}\big(\mathbb{SZ}_\bullet(-), U(1)\big). \tag{4.32}$$

$\mathbb{SZ}$ can be defined as the first Postnikov truncation of the sphere spectrum $\mathbb{S}$ or the Thom spectrum $\mathbb{M}Spin$. Namely, when $Y$ is $(n-1)$-connected, we have

$$\widetilde{\mathbb{SZ}}_\bullet(Y) \simeq \pi_\bullet^s(Y) \simeq \widetilde{\Omega}_\bullet^{Spin}(Y), \qquad \text{for } \bullet \leq n+1. \tag{4.33}$$

Nonzero homotopy groups of the spectra mentioned here are summarized:

| $\mathbb{E}$ | $\mathbb{H}\mathbb{Z}$ | $\mathbb{H}U(1)$ | $\mathbb{SZ}$ | $\mathbb{SU}$ |
|:---:|:---:|:---:|:---:|:---:|
| $\mathbb{E}_1$ | | | $\mathbb{Z}_2$ | |
| $\mathbb{E}_0$ | $\mathbb{Z}$ | $U(1)$ | $\mathbb{Z}$ | $U(1)$ |
| $\mathbb{E}_{-1}$ | | | | $\mathbb{Z}_2$ |

$$\tag{4.34}$$

As a final remark, if we planned to classify gapped phases, we would take the other $(\infty, 1)$-completion at the beginning of Sec. 4.2.2. Then we would reach spectra $\Sigma\mathbb{H}\mathbb{Z}$ and $\Sigma I_{\mathbb{Z}}\mathbb{SZ}$ instead. Here $I_{\mathbb{Z}}\mathbb{SZ}$ denotes the Anderson dual of $\mathbb{SZ}$ and is the spectrum studied by Freed [83] and Gu-Wen [84].

---

[16]This is also the Atiyah–Hirzebruch spectral sequence for $\mathbb{SU}^\bullet(-)$. Namely, $d_2 : E_2^{\bullet,1} \to E_2^{\bullet+2,0} = \rho\circ\mathrm{Sq}^2$.

## 5 Non-invertible structure beyond homotopy groups

Following Ansatz 3.2 and Conjecture 3.3, we have studied the algebraic structure of solitonic symmetry in Sec. 4. These results are pretty formal, and we would like to make some down-to-earth illustrations. For this purpose, we adopt the following angle of looking at solitonic symmetry: What role does the conventional wisdom $\mathrm{Hom}(\pi_\bullet-, U(1))$ (see Sec. 2.2.3) play in the general solitonic symmetry? In this section, we focus on path-connected target space $Y$ since different path components correspond just to different universes[17].

The conventional wisdom classifies topological charges by homotopy groups $\pi_\bullet-$. From the perspective of topological functionals, conventional wisdom concerns topological functionals only on spheres, and solitonic symmetry has the structure $\mathrm{Hom}(\pi_\bullet-, U(1))$. Due to Alexander's trick, a sphere necessarily has

$$\pi_0 \mathrm{Diff}(S^\bullet, \xi) \simeq 0 \tag{5.2}$$

for an orientation $\xi$. Thus spherical topological functionals do not suffer from the identity problem discussed in Sec. 3.1. Therefore, $\mathrm{Hom}(\pi_\bullet-, U(1))$ should not be regarded as just wrong. Instead, it is the precursor of the non-invertible structure.

### 5.1 Rectification vs. condensation

Let us speculate the structure of $\mathsf{Rep}^n(Y)$ and $\mathsf{sRep}^n(Y)$ inductively [We use $\mathsf{Rep}^n(Y)$ to illustrate the idea]. We first suppose that we have already understood $\mathsf{Rep}^{n-1}(Y) \simeq \mathsf{Rep}^{n-1}(\pi_{\leq n-1}Y)$, and we now would like to understand $\mathsf{Rep}^n(Y) \simeq \mathsf{Rep}^n(\pi_{\leq n}Y)$. Recall that $\pi_{\leq\bullet}Y$ denotes $Y$'s homotopy $\bullet$-type, i.e., the $\bullet$-th Postnikov truncation of $Y$. The $n$-th floor of $Y$'s Postnikov tower is a fibration

$$B^n \pi_n Y \to \pi_{\leq n}Y \to \pi_{\leq n-1}Y . \tag{5.3}$$

This fibration allows us to complete our mission in two steps:

$$\mathsf{Rep}^{n-1}(Y) \overset{1}{\Longrightarrow} \mathsf{Rep}^n(\pi_{\leq n-1}Y) \overset{2}{\Longrightarrow} \mathsf{Rep}^n(Y). \tag{5.4}$$

In the first step, we need to construct $n$-dim topological functionals for $\pi_{\leq n-1}Y$ from lower-dim topological functionals. Because $\pi_{\leq n-1}Y$ is $(n-1)$-aspherical, i.e., it has no $n$-dim homotopical data at all, the $n$-dim topological functionals can be obtained via *condensation*.

---

[17]When we are interested in the relationship between different universes, it is also helpful to render multiple path components. In such a case, we can consider a point-like topological functional that specifies one of the path-connected components, i.e., a local operator $1_i(x)$ that takes 1 if the field at $x$ is valued in $i \in \pi_0 Y$ and takes 0 otherwise. Due to the continuity of the field, this operator is indeed topological. It generates the $(d-1)$-form solitonic symmetry,

$$\mathsf{Rep}^0(Y) \simeq \mathsf{sRep}^0(Y) \simeq \left\{\pi_0 Y \mapsto \mathbb{C}\right\}, \tag{5.1}$$

which is non-invertible. The state space decomposes into different sectors, i.e., universes, even at finite volumes [6, 19, 22, 85–90]. Interfaces between different universes, i.e. $(d-1)$-dim solitonic defects, are the charged objects of the $(d-1)$-form solitonic symmetry.

Physically, following Ref. [30], condensation can be formulated via higher-gauging the sub-symmetries of solitonic symmetry on the $n$-dim operator manifold only. Mathematically, following Ref. [68], condensation corresponds to formulating the Karoubi completion. Thus we expect

$$\mathsf{Rep}^n(\pi_{\leq n-1}Y) \;\simeq\; \Sigma\mathsf{Rep}^{n-1}(Y)\,. \tag{5.5}$$

The higher-gauging procedure shows that condensation relies on nontrivial lower-dim cycles on the operator manifold. Therefore, the $n$-dim topological functionals obtained by condensation must take trivial values on $S^n$.

In the second step, the Postnikov fibration (5.3) induces an injective functor

$$\Sigma\mathsf{Rep}^{n-1}(Y) \;\to\; \mathsf{Rep}^n(Y)\,. \tag{5.6}$$

What new objects are added in this step? The homotopy fiber of the fibration (5.3) points out the answer: the topological functionals that are nontrivial on $S^n$. Namely, we expect that the Postnikov fibration induces a "short exact sequence",

$$0 \;\to\; \Sigma\mathsf{Rep}^{n-1}(Y) \;\to\; \mathsf{Rep}^n(Y) \;\to\; \widetilde{\mathsf{Rep}}^n(B^n\pi_nY) \;\to\; 0\,. \tag{5.7}$$

Here the "reduced" solitonic cohomology $\widetilde{\mathsf{Rep}}^n(B^n\pi_nY)$ is the $n$-category completion of the commutative rig, the $\mathbb{N}$-span of $\mathrm{Hom}\big(\pi_nY, U(1)\big)$, by contractible morphisms. This fits our experience with $n$-form gauge fields of gauge group $\pi_nY$ and can be directly verified when $\pi_nY$ is finite. When the Postnikov fibration (5.3) is a trivial product, the "short exact sequence" (5.7) canonically splits, and the conventional wisdom makes the correct prediction. But when the Postnikov fibration (5.3) is nontrivial, the map $\mathsf{Rep}^n(Y) \to \widetilde{\mathsf{Rep}}^n(B^n\pi_nY)$ has a non-invertible kernel, and the fusion rule of spherical topological functionals becomes non-invertible in general. This non-invertible *rectification* of $\mathrm{Hom}\big(\pi_nY, U(1)\big)$ is of our primary interest.

Using the invertible solitonic subsymmetry we discussed in Sec. 4.3.2, we can measure the rectification of $\mathrm{Hom}(\pi_\bullet-, U(1))$ by (generalized) Hurewicz maps,

$$
\begin{aligned}
\text{(bosonic)} \qquad & \pi_\bullet- \;\xrightarrow{h_b}\; H_\bullet(-;\mathbb{Z})\,, \\
\text{(fermionic)} \qquad & \pi_\bullet- \;\xrightarrow{h_f}\; \mathbb{SZ}_\bullet(-)\,.
\end{aligned} \tag{5.8}
$$

Since these homology groups describe the invertible topological charges, the rectification of conventional wisdom is measured by the non-injectivity of Hurewicz maps.

- Two elements in $\pi_\bullet-$ differed by an element in $\ker h_b$ or $\ker h_f$ must share the same invertible topological charge.

- The image of the dual Hurewicz map $H^\bullet\big(-;U(1)\big) \to \mathrm{Hom}\big(\pi_\bullet-, U(1)\big)$ or $\mathbb{SU}^\bullet(-)) \to \mathrm{Hom}\big(\pi_\bullet-, U(1)\big)$ characterizes the survivors in the conventional wisdom as invertible operators.

Recall that the invertible solitonic symmetry we found in Sec. 4.3.2 is probably not maximal. Thus we may overestimate the rectification and regard some invertible operators as non-invertible. Nevertheless, when $Y$ is $(n-1)$-connected, such overestimation does not happen for dimension $\leq n+1$ because our spectra are the $(-1)$-connective covers of the authentic spectra. Besides, the non-rectified operators we find should be truly invertible.

## 5.2 Examples of rectification

The failure of Hurewicz maps' being injective originates from the non-Abelianness in the homotopy structure of the target space $Y$. Namely, we say $Y$ is Abelian if

$$\text{all homotopy groups are Abelian and } Y \simeq \prod_{n=1}^{\infty} B^n \pi_n Y \,.$$

Hurewicz maps on Abelian spaces are always injective, and thus the conventional wisdom $\text{Hom}\big(\pi_\bullet -, U(1)\big)$ is not rectified. However, a general topological space is far from Abelian, and its Postnikov tower describes its non-Abelianness.

### 5.2.1 Spherical rectification

First of all, the homotopy classes of field configurations on a sphere are, in general, not the homotopy group, i.e., $\pi_\bullet Y \not\simeq [S^\bullet, Y]$. This comes from a distinction between homotopies and based homotopies. Let us write based homotopy classes of based maps between two based spaces as $[-,-]_*$. Homotopy groups are precisely $\pi_\bullet Y \equiv [S^\bullet, Y]_*$. Through the homotopy extension property, $\pi_1 Y$ always naturally acts on $[-,Y]_*$. Then $[-,Y]$ is precisely the orbit space of this $\pi_1 Y$-action, i.e. (see Prop. 4A.2 of Ref. [91])

$$[-,Y] \simeq \frac{[-,Y]_*}{\pi_1 Y\text{-action}} \,. \tag{5.9}$$

On homotopy groups $\pi_n Y$, the $\pi_1 Y$-action comprises automorphisms of $\pi_n Y$. As long as $[S^\bullet, Y] \not\simeq \pi_\bullet Y$, the conventional wisdom $\text{Hom}\big(\pi_\bullet -, U(1)\big)$ is rectified.

In the simplest case of $\bullet = 1$, $\pi_1 Y$ acts on itself as inner automorphisms and thus

$$[S^1, Y] \simeq \{\text{conjugacy classes of } \pi_1 Y\} \,, \tag{5.10}$$

We have already seen this around the end of Sec. 4.2.3. The Hurewicz map $\pi_1 Y \to H_1(Y;\mathbb{Z}) \simeq \widetilde{\mathbb{SZ}}_1(Y)$ is just the Abelianization. Therefore, we have

$$H^1\big(Y;U(1)\big) \simeq \text{Hom}\big(\pi_1 Y, U(1)\big) \,, \qquad \mathbb{SU}^1(Y) \simeq \text{Hom}\big(\pi_1 Y, U(1)\big) \times \mathbb{Z}_2 \,, \tag{5.11}$$

which comprises 1-dim (super-)representations of $\pi_1 Y$. The higher-dim (super-)representations of $\pi_1 Y$ constitute the non-invertible objects in $\mathsf{Rep}^1(Y)$ and $\mathsf{sRep}^1(Y)$, the $[\geq (d-2)]$-form solitonic symmetry.

The $\pi_1 Y$-action on higher homotopy groups[18] has no chance to be inner since higher $\pi_\bullet Y$ is Abelian. To describe topological functionals on

$$[S^\bullet, Y] \simeq \pi_\bullet Y \Big/ \pi_1 Y\text{-action} \,, \tag{5.12}$$

we can prescribe a semidirect product from the $\pi_1 Y$-action,

$$\pi_\bullet Y \rtimes \pi_1 Y \,. \tag{5.13}$$

---

[18] There is an illuminating way to understand these actions. $Y$'s universal cover $\bar{Y}$ naturally carries a free $\pi_1 Y$-action. Each $\pi_1 Y$ element as a homeomorphism of $\bar{Y}$ induces a group automorphism on $\pi_n(\bar{Y})$. These automorphisms assemble to a $\pi_1 Y$-action on $\pi_n(\bar{Y})$, which is then turned into a $\pi_1 Y$-action on $\pi_n Y$ by the natural isomorphism $\pi_n(\bar{Y}) \simeq \pi_n Y$ for $n > 1$.

Then a $S^\bullet$ topological functional is a character of $\pi_\bullet Y \rtimes \pi_1 Y$ which vanishes for all group elements outside $\pi_\bullet Y$. A nontrivial $\pi_1 Y$-action implies the existence of such representations of dimension $> 1$ and accordingly, the $S^\bullet$ topological functionals have a non-invertible fusion rule[19]. The simplest example for this effect is perhaps the $S^2$ topological functionals for[20]

$$Y \simeq \mathbb{R}P^2 \simeq S^2/\mathbb{Z}_2 \quad \text{or} \quad Y \simeq BO(2) \simeq BU(1)/\mathbb{Z}_2 \,, \tag{5.14}$$

or any other $Y$ whose $\pi_{\leq 2} Y$ is given by the unique nontrivial split fibration of the following form,

$$B^2\mathbb{Z} \to \pi_{\leq 2}Y \to B\mathbb{Z}_2 \,. \tag{5.15}$$

In these examples, $\pi_1 Y \simeq \mathbb{Z}_2$ acts on $\pi_2 Y \simeq \mathbb{Z}$ through $\mathbb{Z} \to -\mathbb{Z}$, which means

$$[S^2, Y] \simeq \mathbb{N} \,. \tag{5.16}$$

We can find its homology to be

$$H_2(Y;\mathbb{Z}) \simeq 0 \,, \qquad \mathbb{SZ}_2(Y) \simeq 0 \,, \tag{5.17}$$

which implies that the conventional wisdom $\mathrm{Hom}\big(\pi_\bullet -, U(1)\big)$ is completely rectified into non-invertible symmetry. $S^2$ topological functionals are given by characters of $D_\infty \simeq \mathbb{Z} \rtimes \mathbb{Z}_2$ which vanish outside $\mathbb{Z}$. Since there are infinitely many topological charges, according to the discrete approximation, we should consider the representations that are arbitrarily close to representations of quotient $D_n \simeq \mathbb{Z}_n \rtimes \mathbb{Z}_2$ for any finite $n$. Therefore, $S^2$ topological functionals are $\mathbb{N}$-linearly spanned by the characters of 2-dim irreducible representations of $D_\infty$, namely $2\cos(\phi\bullet)$ with $\bullet \in \pi_2 Y \simeq \mathbb{Z}$ for all $\phi \in \mathbb{R}/2\pi\mathbb{Z}$. The non-invertible fusion rule follows evidently,

$$2\cos(\alpha\bullet) \times 2\cos(\beta\bullet) = 2\cos[(\alpha+\beta)\bullet] + 2\cos[(\alpha-\beta)\bullet] \,. \tag{5.18}$$

In summary, the the $\pi_1$-action $\pi_1 \times \pi_\bullet \to \pi_\bullet$ rectifies the conventional wisdom by directly assigning a non-invertible fusion rule to spherical topological functionals.

### 5.2.2 Non-spherical rectification

Let us now discuss subtler effects that cause non-injectivity beyond $\pi_1$-actions. When $Y$ is $(k-1)$-connected with $k > 1$, the Hurewicz map, $h_b : \pi_\bullet Y \to H_\bullet(Y;\mathbb{Z})$, is isomorphism for $\bullet \leq k$ (and same is true for $h_f$), which shows that the solitonic symmetry is invertible at least up to $k$-dim topological functionals. Non-invertibility can appear for larger dimensions, $\bullet \geq k+1$. Since there is no $\pi_1$-action here, topological functionals must be invertible

---

[19]The fact that the $\pi_1$-action can lead to a nontrivial interplay between solitons of different dimensions and an unconventional topological conservation law was also noticed by, e.g., Refs. [92, 93].

[20]The latter example $Y \simeq BO(2)$ was discussed as semi-Abelian gauge theories in Refs. [25, 26, 94]. The non-invertible symmetries for the electric 1-form symmetries have been discussed, while our discussion here is devoted to the magnetic one. In both cases, the mechanism for constructing non-invertible symmetries is essentially the same, and we can be exchanged under a duality transformation. We note a mixed 't Hooft anomaly between the electric and the magnetic symmetry, so we cannot find a path-integral description where both symmetries are solitonic.

as long as they inhabit spheres. Thus rectification happens because a spherical topological functional can also inhabit non-spheres.

To see how the non-invertible symmetry appears in this situation, we need to take into account the inter-dimensional effects between solitons. As we have discussed in Sec. 2.2.3, for some $k \leq p < n$, a codimension-$(p+1)$ solitonic defect may also carry not only the $p$-dim solitonic charge but also the $n$-dim solitonic charge, and such $n$-dim solitonic charge needs to be measured by non-spherical topological functionals. As the Hurewicz map of degree $n$ is not injective, this may violate the selection rule given by $\mathrm{Hom}(\pi_n Y, U(1))$, which suggests that the conventional selection rule, $\mathrm{Hom}(\pi_n Y, U(1))$, is valid only if solitons with higher dimensions are absent and selection rules are modified otherwise. Accordingly, $n$-dim solitonic symmetry has to be non-invertible to capture this intriguing selection rule.

As the present authors pointed out in the previous paper Ref. [1], this effect actually happens in the 4-dim $\mathbb{C}P^1$ sigma model. Let us discuss it here as an example. Homotopy groups of the simply-connected $\mathbb{C}P^1 \simeq S^2$ at low degrees are well-known to be

$$\pi_2(\mathbb{C}P^1) \simeq \mathbb{Z}\,, \qquad \pi_3(\mathbb{C}P^1) \simeq \mathbb{Z}\,. \tag{5.19}$$

The third Postnikov truncation of $\mathbb{C}P^1$ then sits in a principal fibration

$$B^3\mathbb{Z} \;\longrightarrow\; \pi_{\leq 3}\mathbb{C}P^1 \;\longrightarrow\; B^2\mathbb{Z}\,. \tag{5.20}$$

It is well-known that this fibration is classified by a generator of

$$H^4(B^2\mathbb{Z}; \mathbb{Z}) \simeq \mathbb{Z}\,. \tag{5.21}$$

This structure of the homotopy 3-type of $\mathbb{C}P^1$ prescribes the following homologies,

$$H_2(\mathbb{C}P^1; \mathbb{Z}) \simeq \mathbb{Z}\,, \qquad \mathbb{S}\mathbb{Z}_2(\mathbb{C}P^1) \simeq \mathbb{Z}\,, \tag{5.22a}$$

$$H_3(\mathbb{C}P^1; \mathbb{Z}) \simeq 0\,, \qquad \mathbb{S}\mathbb{Z}_3(\mathbb{C}P^1) \simeq \mathbb{Z}_2\,, \tag{5.22b}$$

and requires all the relevant Hurewicz maps to be epimorphic. The last map $\pi_3(\mathbb{C}P^1) \to \mathbb{S}\mathbb{Z}_3(\mathbb{C}P^1)$'s being epimorphic can be understood via the Freudenthal suspension theorem due to the natural $\widetilde{\mathbb{S}\mathbb{Z}}_3(\mathbb{C}P^1) \simeq \pi_3^s(\mathbb{C}P^1)$.

On the one hand, the conventional wisdom is not rectified at dimension 2, which gives a $U(1)$ 1-form symmetry acting on the line solitonic defects. The concrete topological functionals were constructed by Eq. (3.5) in Sec. 3.1. On the other hand, the conventional wisdom is indeed rectified at dimension 3, which gives non-invertible 0-form symmetry acting on point solitonic defects. The extent of rectification depends on the bosonic/fermionic nature of the theory. The conventional wisdom is completely rectified and non-invertible in the bosonic theory, while a $\mathbb{Z}_2$ part survives in the fermionic theory. The concrete topological functional for this $\mathbb{Z}_2$ part was constructed by Eq. (3.9) in Sec. 3.1.

We have also constructed non-invertible 3-dim topological functionals in Ref. [1] according to the discrete approximation. Concretely, we construct the rectified operators for $\mathbb{Z}_{2N} \subseteq \mathrm{Hom}(\pi_3(\mathbb{C}P^1), U(1))$ for each $N \in \mathbb{N}$. In particular, a generator of $\mathbb{Z}_{2N}$ is rectified into the following non-invertible form:

$$\mathcal{H}_{\pi/N}(M) \equiv \int \mathcal{D}b \, \exp\left(-\mathrm{i} \int_M \frac{N}{4\pi} b \, \mathrm{d}b + \mathrm{i} \int_M \frac{1}{2\pi} b \, \mathrm{d}\sigma^* a\right)\,, \tag{5.23}$$

where $\sigma^* a$ is the same as that in Eq. (3.9), i.e., $\sigma|_M : M \mapsto \mathbb{C}P^1$ denotes the field map and $a$ denotes the $U(1)$ gauge field on $S^2$ associated with the Hopf fibration $S^1 \to S^3 \to S^2$. In the fermionic theory, these operators (5.23) are the building block of $\mathsf{sRep}^3(\mathbb{C}P^1)$ beyond condensation $\mathsf{sRep}^3(B^2\mathbb{Z}) \simeq \Sigma\mathsf{sRep}^2(\mathbb{C}P^1)$. In the bosonic theory, $N$ needs to be restricted to even integers without spin structures. These even-$N$ operators are also building blocks of $\mathsf{Rep}^3(\mathbb{C}P^1)$.

When $M_3 = S^3$, these topological functionals recover a naive integral expression for the Hopf invariant,

$$\mathcal{H}_{\pi/N}(S^3) = \frac{1}{\sqrt{N}} \exp\left( i\frac{\pi}{N} \int_{S^3} \frac{\sigma^* a \, d\sigma^* a}{4\pi^2} \right) = \frac{1}{\sqrt{N}} \exp\left( i\frac{\pi}{N} \bullet \right), \qquad (5.24)$$

for $\bullet \in \mathbb{Z} \simeq \pi_3(\mathbb{C}P^1)$, which indeed echoes the conventional wisdom. The non-invertible feature of Eq. (5.23) becomes transparent by evaluating it around the line solitonic defect, which is also the charged object of the $U(1)$ 1-form solitonic symmetry. The line solitonic defect is defined by setting the boundary condition for the $\mathbb{C}P^1$ field on the normal sphere bundle around $S^1$, which is nothing but $S^2 \times S^1$. Based on to the structure of $[S^2 \times S^1, S^2]$ described by Eq. (3.6), for $(m, \ell) \in [S^2 \times S^1, S^2]$, we can evaluate to have

$$\mathcal{H}_{\pi/N}(S^2 \times S^1) = \begin{cases} \exp\left( i\frac{\pi}{N}\ell \right), & m = 0 \mod N \\ 0, & m \neq 0 \mod N \end{cases} . \qquad (5.25)$$

It does not lift the $g$-degeneracy discussed below Eq. (3.7) as we expected in Sec. 3.1. We see that the topological functional becomes nonzero only if the level $N$ divides the 1-form topological charge $m$, which clarifies that the presence of line solitonic defects causes the non-invertibility of the 0-form solitonic symmetry. Also, we see that different manifolds can support coherent topological charges like that $\ell$ is correlated to $\pi_3(\mathbb{C}P^1)$.

## 5.3 Examples of condensation

Finally, we describe some examples of topological functionals that are trivial on spheres, i.e., those that lie in the image of Eq. (5.6). In fact, Sec. 5.2.2 already implies examples when we try to fusion some operators there. However, here we shall present several more clean examples where conventional wisdom totally vanishes. We focus on the simplest 2-dim case, so suppose $M$ is a closed 2-dim manifold.

Prototypical bosonic examples are the 2-dim topological functionals for two $\mathbb{Z}_n$ gauge fields $s_{1,2} \in H^1(M; \mathbb{Z}_n)$, or for two $S^1$-valued scalars $\phi_{1,2} : M \mapsto \mathbb{R}/2\pi\mathbb{Z}$. The former case has $Y \simeq B\mathbb{Z}_n^2$ while the latter has $Y \simeq T^2$. We see always that $\pi_2 Y \simeq 0$ but

$$H^2\left(B\mathbb{Z}_n^2; U(1)\right) \simeq \mathbb{Z}_n \quad \text{and} \quad H^2\left(T^2; U(1)\right) \simeq U(1). \qquad (5.26)$$

The corresponding 2-dim bosonic topological functionals inhabit Riemann surfaces $M$. They are trivial on $M \simeq S^2$ but non-trivial on higher-genus surfaces such as $M \simeq T^2$. These topological functionals can be explicitly constructed via the universal invertible form (2.7). For $Y \simeq B\mathbb{Z}_n$, we have

$$U_k(M) \equiv \exp\left\{ ik\frac{2\pi}{n} \int_M s_1 \cup s_2 \right\}, \qquad k \in \mathbb{Z}, \, k \sim k + n. \qquad (5.27)$$

For $Y \simeq S^1$, we have

$$U_\theta(M) \equiv \exp\left\{ \mathrm{i}\theta \int_M \frac{\mathrm{d}\phi_1}{2\pi} \wedge \frac{\mathrm{d}\phi_2}{2\pi} \right\}, \qquad \theta \sim \theta + 2\pi. \tag{5.28}$$

They are apparently trivial on $S^2$. They can be defined just via field configurations on several properly selected loops inside $M$.

Prototypical fermionic examples are the 2-dim topological functionals for a $\mathbb{Z}_{2n}$ gauge field $s \in H^1(M; \mathbb{Z}_{2n})$, or for a $S^1$-valued scalar $\phi : M \mapsto \mathbb{R}/2\pi\mathbb{Z}$. The former has $Y \simeq B\mathbb{Z}_{2n}$ while the later has $Y \simeq S^1$. We can readily see $\pi_2 Y \simeq H_2(Y; \mathbb{Z}) \simeq 0$ and find via the long exact sequence (4.31) that

$$\mathbb{S}\mathbb{Z}_2(Y) \simeq H_1(Y; \mathbb{Z}_2) \simeq \mathbb{Z}_2. \tag{5.29}$$

We thus see that the corresponding topological functionals are fermionic, i.e., they inhabit spin Riemann surfaces $M$. Also, they are trivial on $S^2$ but non-trivial on other spin Riemann surfaces. Via the universal invertible form (2.7), we can construct these fermionic topological functionals through the Arf invariant of spin structures. Namely, we pick up a spin structure $\rho$ on $M$ such that $\mathrm{Arf}(\rho) = 0$, i.e., $[\rho] \in 0 \in \Omega_2^{Spin}$. Recall that, on top of an orientation, spin structures form an $H^2(M; \mathbb{Z}_2)$-torsor. Then for $Y \simeq B\mathbb{Z}_{2n}$, we have

$$U_k(M) \equiv (-)^{k\,\mathrm{Arf}\left(\rho + \bar{s}\right)}, \qquad k \in \mathbb{Z}, \; k \sim k + 2, \tag{5.30}$$

where $\bar{s}$ denotes the mod-2 reduction of $s$. For $Y \simeq S^1$, we just substitute $\bar{s}$ above with the mod-2 reduction of $[\phi] \in H^2(M; \mathbb{Z})$. It is evident to see that this topological functional vanishes on $S^2$.

We now carefully analyze the identity problem (see Sec. 3.1) of the above example. Concretely, we consider torus topological functionals for a $S^1$-valued scalar or a $\mathbb{Z}_n$-valued gauge field. Recall that the target space takes the form $Y \simeq BR$ for $R \simeq \mathbb{Z}$ in the former case and for $R \simeq \mathbb{Z}_n$ in the latter case. We readily recognize

$$[T^2, BR] \simeq H^1(T^2; R) \simeq R^2. \tag{5.31}$$

The mapping class groups of $T^2$ are well-known, e.g.,

$$\pi_0 \mathrm{Diff}(T^2) \simeq GL(2, \mathbb{Z}), \tag{5.32a}$$

$$\pi_0 \mathrm{Diff}(T^2, \xi) \simeq SL(2, \mathbb{Z}), \tag{5.32b}$$

$$\pi_0 \mathrm{Diff}(T^2, \xi, \rho) \simeq p^{-1}\left(\mathbb{Z}_2\right), \quad \text{for } \mathbb{Z}_2 \subseteq SL(2, \mathbb{Z}_2) \text{ and } SL(2, \mathbb{Z}) \xrightarrow{p} SL(2, \mathbb{Z}_2). \tag{5.32c}$$

where $\xi$ is an orientation and $\rho$ is still a spin structure such that $\mathrm{Arf}(\rho) = 0$. Without loss of generality, $\rho$ can be taken as antiperiodic-periodic. Since $[T^2, BR]$ is just a 2-dim lattice, these mapping class groups act on $[T^2, BR]$ just as 2-by-2 $\mathbb{Z}$-valued matrices. These actions are far from trivial, and concretely, the criteria for classifying $(a, b) \in [T^2, BR]$ into its action orbit are as follows:

$$\pi_0 \mathrm{Diff}(T^2): \; \gcd(a, b), \tag{5.33a}$$

$$\pi_0 \mathrm{Diff}(T^2, \xi): \; \gcd(a, b), \tag{5.33b}$$

$$\pi_0 \mathrm{Diff}(T^2, \xi, \rho): \; \gcd(a, b), \; a \bmod 2, \tag{5.33c}$$

where we stipulate $\gcd(0,0) \equiv 0$. We see that a spin structure slightly lifts the degeneracy when $R$ has an even characteristic. This tiny degeneracy lifting is exactly realized by the invertible topological functional (5.30). Despite huge degeneracy, equation (5.33) still gives tremendous inequivalent topological charges, and they have to be distinguished by non-invertible topological functionals. They are examples of non-invertible condensations.

# 6 Discussion

In this paper, we discussed the general structure of the non-invertible solitonic symmetry by attempting a precise mathematical formulation (Sec. 4) after a pursuit of the proper physical constraints (Sec. 3). It allows us to discuss how the non-invertible structure in general solitonic symmetry goes beyond the conventional wisdom of homotopy groups (Sec. 5) in terms of concrete examples. Nevertheless, our analysis in this paper is still far from complete to be complemented by future studies. In this section, we are going to make some remarks and discuss the outlooks for future studies.

## 6.1 Remarks

First, we point out an interesting connection between the solitonic symmetry and the higher-group symmetry, which was actually already mentioned around the end of Sec. 2.1. Assume we have a QFT $\mathcal{T}$ with an anomaly-free discrete higher-group $\mathsf{G}$. Then we can consider another QFT $\mathcal{T}/\mathsf{G}$ by dynamically gauging the higher-group $\mathsf{G}$. This gauging procedure can be realized by adding a path integral over $\mathsf{G}$ gauge fields. The homotopy target space of this path integral is precisely the classifying space $B\mathsf{G}$. Thus the theory $\mathcal{T}/\mathsf{G}$ acquires the solitonic symmetry $(\mathsf{s})\mathsf{Rep}^\bullet(B\mathsf{G})$. Therefore, gauging a non-anomalous higher-group symmetry produces a QFT with non-invertible solitonic symmetry. In particular, this picture suggests the last property of solitonic symmetry described in this paper.

**Conjecture 6.1** *Solitonic symmetry is free of 't Hooft anomaly.*

It comes from the belief that the dynamical gauging procedure is reversible. Namely, if we could develop the proper gauging procedures of the solitonic symmetry, then it would be natural to expect that gauging of $(\mathsf{s})\mathsf{Rep}^\bullet(B\mathsf{G})$ in $\mathcal{T}/\mathsf{G}$ produces the original theory $\mathcal{T}$. This picture echoes the ideas developed in Refs. [52–55]. One may consider a Tannaka duality between higher-group symmetry and solitonic symmetry. This also suggests a wide adaption of solitonic symmetry: A broad class of non-solitonic symmetry can also be described by solitonic symmetry via a properly selected virtual target space $Y$.

Second, we point out that QFTs with discrete higher-group symmetry also possess solitonic sectors. Their solitonic sectors are the higher-domain-walls of spontaneously breaking higher-group symmetry and inhabit only non-closed manifolds (recall Footnote 5). However, via the dynamical gauging discussed above, these non-closed solitonic sectors are the counterparts of our closed solitonic sectors addressed in this paper. For example, when a discrete symmetry is spontaneously broken, there are domain walls connecting different vacua, and they are well-defined on the infinite space. However, when we consider the closed space with the periodic boundary condition, a single domain wall is inconsistent with

the boundary condition. Even in this case, the single domain wall becomes well-defined on closed manifolds by considering the gauging of the broken symmetry or the symmetry-twisted sector. Therefore, solitons in different theories can behave locally similarly but have different global behaviors.

## 6.2 Outlooks

We present a list of outlooks based on the subtle points in the analysis of fully-extended TQFTs in the present paper. They should also be of interest to those concerned about classifying gapped phases.

- Does the Karoubi completeness, as well as the operation $\Sigma$, provide a fully satisfactory solution to the codomains for fully-extended TQFTs? Does $\Sigma$ correctly capture the physical condensation?

- The validity of Conjecture 4.4 on the homotopy actions should be confirmed.

- The maximal invertible solitonic symmetry, i.e., the structure of the infinite loop spaces $\left(\Sigma^{n-1}\mathsf{Vect}^{fd}\right)^{\times}$ and $\left(\Sigma^{n-1}\mathsf{sVect}^{fd}\right)^{\times}$ for $n > 1$, should be clarified. The higher-group structure of invertible solitonic symmetry should also be clarified.

As for the solitonic symmetry itself, we also have many prospects, such as exploring the relation between solitonic symmetry and other generalized symmetry, constructing explicitly more examples of topological functionals, and presenting a comprehensive classification of low-dim topological functionals. A list of intriguing general questions is as follows.

- The validity of Conjecture 3.3 on the correspondence between TQFTs and their partition functions should be confirmed.

- A systematic treatment of continuous solitonic symmetry and infinitely many topological charges should be developed. A proposal is to consider things like $\Sigma^{n-1}\mathsf{Hilb}^{fd}$ (c.f. Ref. [81]). Another is to make the discrete approximation rigorous.

- We should rigorously formulate the inductive decomposition into non-spherical condensation and spherical rectification discussed in Sec. 5.1. Namely, we should prove our expected "short exact sequence" (5.7) and determines its structure by the Postnikov fibration.

- Can we tackle solitonic symmetry from the side of topological charges? This requires a complicated analysis of the homotopy classes of maps on manifolds, including their relations via bordisms (see Ref. [1]) to resolve the coherence problem (see Sec. 3.2), as well as a direct treatment of the identity problem (see Sec. 3.1).

If these problems can be solved, we will further strengthen our confidence in Ansatz 3.2 or know how to modify it. This work thus serves as an outset in the pursuit of a perfect description of solitonic symmetry.

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
