# Peer review of "Solitonic symmetry as non-invertible symmetry: cohomology theories with TQFT coefficients"

_SciPost Physics_

## Round 2 · Referee Report · Anonymous (Referee 1) · 2024-6-4

Strengths

This is a timely paper on noninvertible symmetries, following up a short preliminary note written by the same authors. This paper gives a much more in-depth treatment of solitonic symmetries generalizing "Hopfions" of 3d sigma models with target CP^1.

Weaknesses

The paper uses some rather technical mathematics (eg spectra in the sense of stable homotopy theory) that will not be familiar to most readers; however, this is an unavoidable aspect of their construction. To be clear, I have not checked the details of their homotopy computations, but no errors jumped out at me.

Report

Briefly, this is a timely and important paper, that is deserving of publication, at least after fixing a handful of minor typos.

Requested changes

I noticed a small number of typos.
(1) p 2, "gepped" should be "gapped"
(2) p 6, equ'n (2.8), in the second line, is it an n-dimensional or (n+1)-dimensional topological function in (d+1)-dim QFT ?
(3) p 12, second paragraph of section 3.2: "E-orientable E-oriented" seems slightly redundant; unless I'm misunderstanding the terminology, saying "E-oriented" would ordinarily imply "E-orientable," and so would suffice.
(4) p 25, last line: "unifromized" should be "uniformized"

Recommendation

Ask for minor revision

---

## Round 2 · Referee Report · Anonymous (Referee 2) · 2024-6-7

Strengths

- Ambitious aim to establish a general mathematical framework dealing with general cases of generalized symmetry.
- Connecting the physics context to profound and rich math results.
- Original ideas.

Weaknesses

- Not easy to read for those who are not familiar with relevant math literature.

Report

The manuscript concerns the symmetry of quantum field theory. It focus on what they call the "solitonic symmetry", whose existence depends only on the structure of the field contents, and not on the action.
The manuscript proposes to use the mathematical framework of functorial fully-extended TQFT, founded by Lurie, to describe such generalized symmetry. The manuscript is a very ambitious, and
connects the physics context to profound and rich math results. Thus I recommend to publish it in SciPost. However, I have a few (major-ish) points that I want to be clarified, and other minor points to be considered, before publishing the manuscript.

Requested changes

# Major Points
1. Structure for submanifold
- Throughout the paper, the topological functional on a submanifold $M$ of a spacetime is considered. The author assumes that such fuctional can depend on an orientation of $M$ for a bosonic theory, or a spin-structure of $M$ for a fermionic theory.
- However, in general one cannot canonically induce such a structure onto $M$ using the corresponding structure in the ambient space. For example, there are unorientable submanifolds in an oriented manifold. Even when $M$ is orientable, its orientation is not cannonically determined by the orientation of the ambient manifold.
- Thus, I want the following points be clear: how is the (spin/oriented) structure on $M$ is supposed to be chosen? Can we also consider unorientable/non-spinnable $M$ to support the operator?

2. Chern-Simons v.s. nVect
- It is known that 3Vect does not include general Cern-Simons theory. See Corollary 2 (page 5) and around of https://arxiv.org/abs/1312.7188 . About that what is considered in 1312.7188 is $\Sigma^2 Vect$, see Conjecture 3.3.5 of Gaiotto-Johnson-Freyd https://arxiv.org/pdf/1905.09566 .
- Physically, $\Sigma^2 Vect$ only includes the TQFTs obtained by gauging (potentially non-invertible) finite symmetry acting on the trivial theory, and a general Chern-Simons theory is not of that type because a finite gauging cannot change chiral central charge.
- Therefore the main example (5.23) cannot be captured by the choice of the target the authors (or the current manuscript) made.
- There are variants of (5.23) that can be constructed using $\Sigma^2 Vect$, however, resembling the construction in C.2 of https://arxiv.org/abs/2205.05086 . Consider to replace the example to the variant.

3. About invertible part (Section 4.3)
- The (supposedly "universal") target for invertible TQFT is proposed in Freed-Hopkins https://arxiv.org/pdf/1604.06527, which is the Brown-Comenetz dualizing spectrum $I_\mathbb{C^\times}$ and Anderson dualizing spectrum $I_Z$. The former is conjectured to classify the invertible theories, while the latter is conjectured to classify the deformation classes. The Freed's $E$-theory in [83] and Gu-Wen cohomology are 1-trucation of $I_Z MTspin(n) \cong I_Z Mspin$.
- Note that Freed-Hopkins also uses discrete topology on $\mathbb{C}$, so I do not think the choice listed in page 19 explains the discrepancy among $I_Z, I_{\mathbb{C}^\times}$, $\Sigma^\bullet Vect^{\times}$. I want clarification on this point.
- Also the discussion in page 28 (in particular (4.32) ) of the current manuscript suggesting $\mathbb{SU}$ is indeed $I_\mathbb{C^\times} MSpin$ at 1-truncation. If this is correct, clear reference to Freed-Hopkins is needed.

# Minor Points

1. "Coleman etc" in page 3. :
The authors could show more respect for Coleman and his contemporaries by spending more words than "etc".

2. ”Then this topological functional can produce a nontrivial number as long as the field configuration on its supporting manifold cannot be continuously deformed to a trivial configuration." in page 7:
This sentence needs clarifications:
- "continuously deformed": what kind of operation is allowed? In this sentence, "collapsing the defect to a point when its worldvolume is bounded within the ambient manifold" be counted as continuous deformation?
- For a non-invertible defect, the functional can be other than 1, even when its worldvolume is a sphere.
- contextually, probably "as long as" would be replaced by "only if".

3. "conventional wisdom" in page 8:
The authors can cite some conventional references crucial to the topic.

4. In a bosonic (resp. fermionic) theory, topological functionals are supposed to inhabit oriented (resp. spin) manifolds.:
Refer to major point 1.

5. Definition 3.1 in page 13:
A conventional definition of the term $n$-finite space in the relevant literature includes $(n+1)$-asphericity . E.g. https://arxiv.org/abs/1703.09764 . Better to note the discrepancy.

6. "Different models for organizing this have been proposed, and the equivalence between them, though widely believed, is a matter of ongoing research.":
Equivalence among many of such models are claimed to be proven in https://doi.org/10.1090/jams/972 .

7. "π0Y worth of universes" in page 18:
footnote 17, or the latter part of it, might be migrated into here.

8. "It is reasonable to anticipate that having bosonic (resp. fermionic) state spaces logically implies inhabiting any oriented (resp. spin) spacetime, " in page 21:
a general Chern-Simons theory is a famous counterexample to this anticipation. This conjecture is more specific to the particular target of iterated Karoubi completion, or finite gauging.

9. "This result hints at the physical meaning of other morphisms in Repn(Y ) and sRepn(Y )":
This assertion is originally from Lurie [64], Section 4.3. See also https://arxiv.org/pdf/2403.01651, on top of page 13, for a more brief explanation. It would be nice if authors can connect their observation with the original stratified cobordism hypothesis.

10. Prop 4.11, "defined via (4.28)" in page 27
It is hard to read off what this means. Please write the definition explicitly.

11. "a spherical topological functional can also inhabit non-spheres" in page 33:
Probably "a topological functional can also inhabit non-spheres".

12. "It is well-known that this fibration is classified" in page 33:
This is directly from the (homotopy) fiber sequence and thus better to name it.

13. Section 5.3 :
It would be better also cite references that discusses relevant or similar examples of condensation defects.

14. Conjecture 6.1:
This conjecture seems to rely on the fact that finite gauging is reversible. However the solitonic symmetry in general comes from continuous gauging/dynamical field, e.g. CP^1 sigma model, which does not seem to admit "ungauging" to the trivial theory. Do the authors expect the conjecture still holds for such cases, and if so what is the rationale?

Recommendation

Ask for major revision

---

## Editorial Decision

resubmitted